# TopBP1 utilises a bipartite GINS binding mode to support genome replication

Matthew Day [1,2,10] ✉, Bilal Tetik [3,10], Milena Parlak [3,10],
Yasser Almeida-Hernández [4,5], Markus Räschle [6], Farnusch Kaschani [7,8],
Heike Siegert [3], Anika Marko [3], Elsa Sanchez-Garcia [4,5], Markus Kaiser [7,8],
Isabel A. Barker [2], Laurence H. Pearl [2,9] ✉, Antony W. Oliver [2] ✉ &
Dominik Boos [3] ✉

Activation of the replicative Mcm2-7 helicase by loading GINS and Cdc45 is crucial for replication origin firing, and as such for faithful genetic inheritance. Our biochemical and structural studies demonstrate that the helicase activator GINS interacts with TopBP1 through two separate binding surfaces, the first involving a stretch of highly conserved amino acids in the TopBP1-GINI region, the second a surface on TopBP1-BRCT4. The two surfaces bind to opposite ends of the A domain of the GINS subunit Psf1. Mutation analysis reveals that either surface is individually able to support TopBP1-GINS interaction, albeit with reduced affinity. Consistently, either surface is sufficient for replication origin firing in *Xenopus* egg extracts and becomes essential in the absence of the other. The TopBP1-GINS interaction appears sterically incompatible with simultaneous binding of DNA polymerase epsilon (Polε) to GINS when bound to Mcm2-7-Cdc45, although TopBP1-BRCT4 and the Polε subunit PolE2 show only partial competitivity in binding to Psf1. Our TopBP1-GINS model improves the understanding of the recently characterised metazoan pre-loading complex. It further predicts the coordination of three molecular origin firing processes, DNA polymerase epsilon arrival, TopBP1 ejection and GINS integration into Mcm2-7-Cdc45.

Duplicating the genome accurately and exactly once each cell cycle lies at the heart of faithful genetic inheritance. The effective execution and regulation of DNA replication origin firing is pivotal to accurate genome duplication. Our understanding of how origin firing in eukaryotes generates replisomes from their inactive precursors, pre-replicative complexes that comprise double hexamers of the Mcm2-7 helicase (MCM-DH), has increased. However, how the two helicase activators GINS and Cdc45 are loaded onto MCM-DHs remains less well understood.

[1]School of Biological and Behavioural Sciences, Blizard Institute, Queen Mary University of London, London E1 2AT, UK. [2]Cancer Research UK DNA Repair Enzymes Group, Genome Damage and Stability Centre, School of Life Sciences, University of Sussex, Falmer, Brighton BN1 9RQ, UK. [3]Molecular Genetics II, Center of Medical Biotechnology, University of Duisburg-Essen, Universitätsstraße 2-5, 45141 Essen, Germany. [4]Computational Bioengineering, Fakultät Bio- und Chemieingenieurwesen, Technical University Dortmund, Emil-Figge Str. 66, 44227 Dortmund, Germany. [5]Computational Biochemistry, Center of Medical Biotechnology, University of Duisburg-Essen, Universitätsstraße 2-5, 45141 Essen, Germany. [6]Molecular Genetics, Technical University Kaiserslautern, Paul-Ehrlich Straße 24, 67663 Kaiserslautern, Germany. [7]Analytics Core Facility Essen, Center of Medical Biotechnology, University of Duisburg-Essen, Universitätsstraße 2-5, 45141 Essen, Germany. [8]Chemical Biology, Center of Medical Biotechnology, University Duisburg-Essen, Fakultät Biologie, Essen, Germany. [9]Division of Structural Biology, Institute of Cancer Research, Chester Beatty Laboratories, 237 Fulham Road, London SW1E 6BT, UK. [10]These authors contributed equally: Matthew Day, Bilal Tetik, Milena Parlak. ✉e-mail: matthew.day@qmul.ac.uk; Laurence.Pearl@sussex.ac.uk; Antony.Oliver@sussex.ac.uk; dominik.boos@uni-due.de

MCM-DHs form during origin licensing in the G1 phase of the cell cycle when the activity level of cyclin-dependent kinases (CDKs) are low[1]. A set of essential licensing factors sequentially loads two Mcm2-7 helicase hexamers onto origin DNA. Double-stranded DNA passes through the central channels of these helicase-inactive MCM-DHs[2–4]. In the following S phase, high CDK activity induces origin firing, converting each MCM-DH into two replisomes travelling in opposite directions. During this process, the MCM-DH separates, helicase activity is switched on, the dsDNA running inside the hexamers is unwound, and what will become the single lagging strand DNA template is excluded from each hexamer (Supplementary Fig. 1). The two active helicases pass each other, engaging with the DNA fork with their respective N-terminal faces first[5,6].

Mcm2-7 helicase activation involves the loading of Cdc45 and GINS onto MCM-DHs to form the CMG (Cdc45-Mcm2-7-GINS) helicase[7–10]. CMG assembly occurs tightly coupled with partial Mcm2-7 double hexamer splitting and limited untwisting of the dsDNA inside the central CMG channels[6,11,12]. The subsequent assembly of mature replisomes involves several steps and many additional replisome components[6,10].

In budding yeast, Cdc45 and GINS integration requires Dbf4-dependent kinase (DDK), S-CDK, the Sld3-Sld7 protein complex (Treslin-MTBP in higher eukaryotes), Dpb11 (TopBP1), Sld2 (potential human equivalents RecQL4 and DONSON) and DNA polymerase epsilon (DNA Polε)[10,13,14]. DDK binds MCM-DHs and phosphorylates the Mcm2-7 helicase, allowing Sld3-Sld7$^{Treslin-MTBP}$ association[15–17]. MCM-DH-bound Sld3-Sld7$^{Treslin-MTBP}$ recruits Cdc45 involving a direct Sld3$^{Treslin}$-Cdc45 interaction[14,18,19]. Sld3-Sld7$^{Treslin-MTBP}$ is also involved in recruiting Dpb11$^{TopBP1}$. For this, S-CDK phosphorylates Sld3$^{Treslin}$ on conserved CDK consensus sites, which can then bind to the N-terminal BRCT (breast cancer type 1 susceptibility) domains of Dpb11$^{TopBP1}$[20–22] (Fig. 1a(i)). The interaction between Sld3$^{Treslin}$ and Dpb11$^{TopBP1}$ serves to recruit the pre-loading complex (pre-LC) to MCM-DHs, comprising Dpb11$^{TopBP1}$, GINS, Sld2$^{RecQL4/DONSON}$ and DNA Polε[23]. Pre-LC integrity requires a CDK phosphorylation-mediated interaction between phosphorylated Sld2$^{RecQL4/DONSON}$ and the C-terminal BRCT-pair of Dpb11$^{TopBP1}$ (Fig. 1a(i)). Sld2$^{RecQL4/DONSON}$, Dpb11$^{TopBP1}$ and Sld3-Sld7$^{Treslin-MTBP}$ then detach from the origin-bound Mcm2-7[13,24], leaving GINS and Cdc45 behind to form the CMG helicase. DNA Polε associates with CMG in a dynamic fashion to form the CMGE complex (CMG-DNA Polε)[25]. The N-terminus of the Dpb2 subunit of DNA Polε and its vertebrate equivalent PolE2 bind to the Psf1 subunit of GINS[26–28]. It has been suggested that this interaction represents the essential activity of DNA Polε in replication origin firing[25].

The precise molecular mechanism underpinning how the metazoan TopBP1$^{Dpb11}$, Treslin$^{Sld3}$ and MTBP$^{Sld7}$ deliver GINS and Cdc45 to the Mcm2-7 helicase remains unclear. TopBP1$^{Dpb11}$ is a multi-BRCT domain scaffold protein containing nine BRCT domains[29]. The triple-BRCT module at the N-terminus of TopBP1$^{Dpb11}$ plus the BRCT4/5 pair share homology with budding yeast Dpb11 (Fig. 1a(i))[30]. As Dpb11, TopBP1 facilitates the coupling of origin firing to the cell cycle using its N-terminal BRCT modules[21,22,31]. TopBP1$^{Dpb11}$-BRCT4/5 was, however, found to be non-essential to replication in Xenopus egg extracts[31]; perhaps surprisingly as the equivalent BRCT module of Dpb11$^{TopBP1}$ (BRCT3/4) is essential[23]. The minimal fragment of TopBP1$^{Dpb11}$ that supports replication includes BRCT0/1/2, BRCT3 (not conserved in Dpb11) and a short segment C-terminal to BRCT3 termed the GINS interaction or GINI region[31,32]. Yeast-two hybrid experiments showed that the GINI regions of TopBP1 and Dpb11, which have little sequence identity, both bind to GINS[32] (Fig. 1a(i)). As both TopBP1 and Dpb11 are limiting factors for origin firing[33–35], it is likely that they are released and recycled to facilitate replication from multiple origins[36]. Both association and disassociation of Dbp11$^{TopBP1}$ are therefore important processes.

Recent evidence showed that TopBP1$^{Dpb11}$ and GINS are members of a metazoan pre-LC-like complex that also contains DNA Polε and a new metazoa-specific origin firing factor, DONSON[37–41]. Pre-LC integrity appears to be important for metazoan origin firing because the interactions of DONSON with TopBP1$^{Dpb11}$ (via the essential TopBP1$^{Dpb11}$-BRCT3 domain), GINS and Mcm3 are necessary for efficient origin firing. The TopBP1$^{Dpb11}$-GINS interaction was not characterised in detail in these recent publications.

Here, we have used purified proteins to address how TopBP1$^{Dpb11}$ loads GINS onto MCM-DHs to activate the Mcm2-7 helicase during replication origin firing. Our structural, biochemical, and functional studies uncover the binding mode of GINS to TopBP1$^{Dpb11}$ and reveal a role for the TopBP1$^{Dpb11}$-BRCT4 domain in replication, previously thought to be dispensable. Our data provide a model and framework for future investigation of CMG formation through GINS and Cdc45 loading by TopBP1$^{Dpb11}$ and the Treslin-MTBP$^{Sld3-Sld7}$ complex. Moreover, our study provides insight how TopBP1$^{Dpb11}$ acts at a molecular level. Such detailed understanding of TopBP1$^{Dpb11}$ is key to unravelling how this important scaffold protein can integrate its many different cellular roles in chromosome biology[29].

## Results

### TopBP1 interacts with GINS in vivo and in vitro

As previous yeast two-hybrid experiments showed an interaction of the human and Xenopus TopBP1-GINI region with GINS (Psf1 and Psf3)[32], we tested if a similar interaction could be detected in human cells. Three proximity biotinylation experiments using cells expressing APEX2-tagged TopBP1 resulted in a specific enrichment of Psf3 upon streptavidin pulldown and mass spectrometry, amongst a range of other known TopBP1 interactors (Supplementary Fig. 2a). Psf2 was also detected in one experiment. Immunoprecipitation from lysates of cells transfected with TopBP1-BRCT0-5 (aa, amino acids 1–766) showed a weak but specific signal for the GINS subunit Sld5 (Supplementary Fig. 2b, c). These experiments suggest that TopBP1 and GINS interact in human cells.

In addition, purified recombinant TopBP1-BRCT0-5 (TopBP1-0-5-WT) co-eluted with the GINS complex from a size exclusion column (Fig. 1b; Supplementary Fig. 3a). These results demonstrate that a TopBP1 fragment spanning the BRCT repeats 0-5 is sufficient to interact with GINS both in vivo and in vitro.

### Two binding motifs within TopBP1 contribute to GINS binding

We then used deletion and mutation analysis to narrow down the GINS binding region within the TopBP1-BRCT0-5 fragment. Removing the Treslin-binding BRCT0-2 module (TopBP1-ΔBRCT0/1/2) or BRCT3 (TopBP1-ΔBRCT3) from TopBP1 (Fig. 1a(ii)) did not affect binding (Supplementary Fig. 3b,c). In contrast, deleting BRCT4/5 downstream of the GINI region (TopBP1-ΔBRCT4/5) strongly impaired GINS interaction (Fig. 1c). We confirmed that the phospho-binding pocket of BRCT5 was not required for this interaction, using a triple point mutant that compromises its phospho-binding ability (B5mut)[42] (Fig. 1a(ii), Supplementary Fig. 3d,e)). Our observations strongly suggested that a second GINS-binding motif, in addition to the GINI region, is present in the BRCT4/5 module. This was interesting, as no replication role for TopBP1-BRCT4/5 had been defined previously.

### Cryo-EM structure of TopBP1-BRCT4/5 in complex with GINS

To reveal how TopBP1 interacts with the GINS complex, peak fractions of TopBP1-BRCTΔ0/1/2 in complex with GINS eluting from the size exclusion column (Supplementary Fig. 3b) were used for cryo-EM. Movies were collected on an FEI Titan Krios microscope equipped with a Gatan K3 camera. 18,945 movies were processed using a combination of CryoSPARC[43] and RELION[44] (Supplementary Fig. 4, Supplementary Table 1) to produce a final 3-dimensional reconstruction, derived from 208,115 particles at 4.6 Å based on Fourier shell correlation (FSC) values. Density corresponding to the GINS tetramer could be readily identified in the resultant map, allowing facile placement of an X-ray

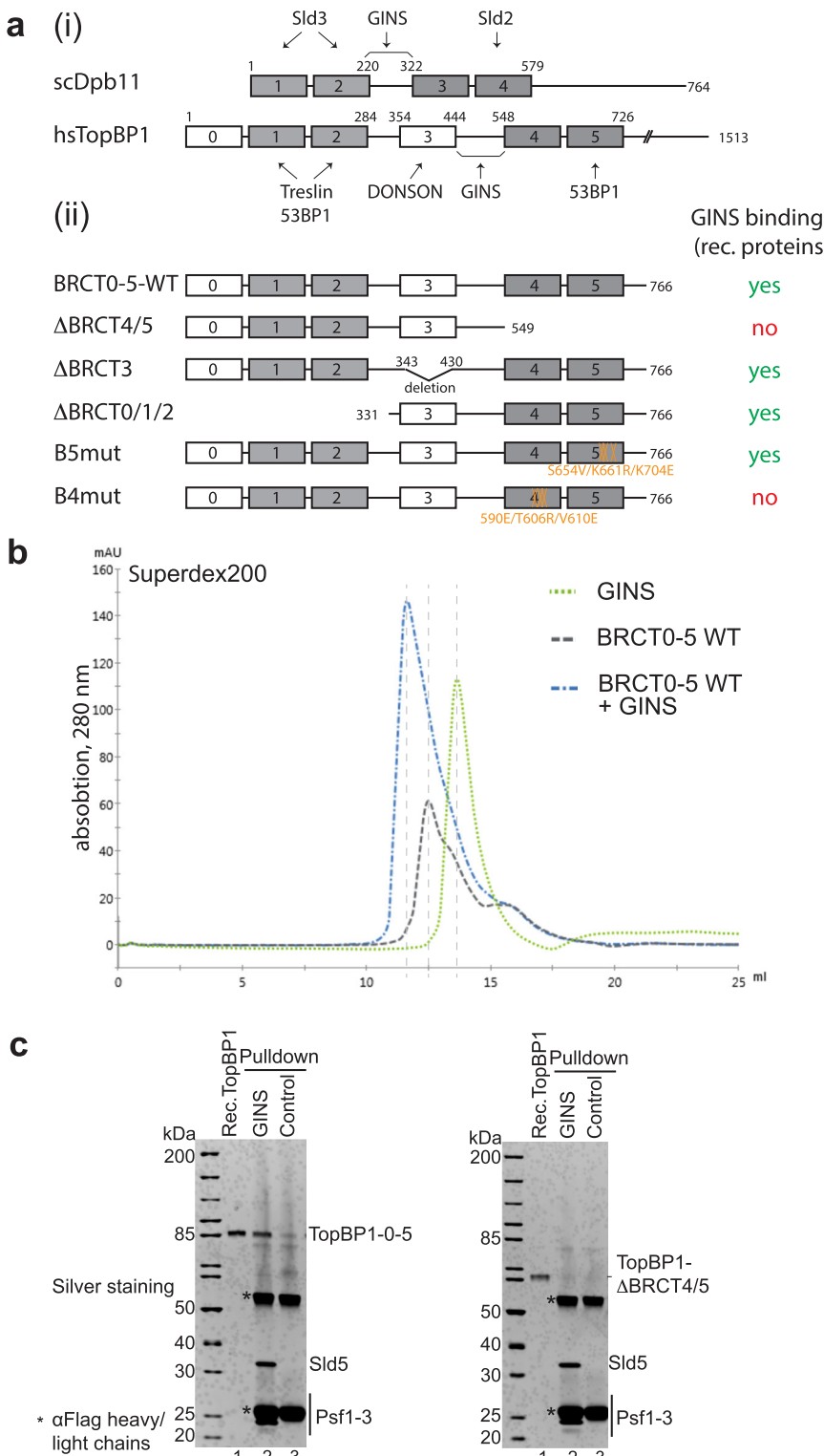

**Fig. 1 | GINS and TopBP1 interact. a** (i) Domain model of human (hs) TopBP1, its conservation with BRCT1/2 and 3/4 domains (grey boxes) of budding yeast Dpb11 (scDpb11). Arrows point to binding domains for interacting proteins. (ii) Overview of TopBP1 wild type (WT) and BRCT deletion (Δ) and point mutants. GINS binding capabilities are labelled in red and green. Numbers, amino acid position; capital letters indicate amino acid substitutions. Amino acid substitutions of point mutants in BRCT4 (B4mut) and 5 (B5mut) are coloured orange. **b** Elution profiles of size

exclusion chromatography (Superdex200) of individual recombinant TopBP1-BRCT0-5 protein (grey) and GINS tetramers (green), or of both upon incubation (blue). **c** Recombinant GINS immobilised on Flag-beads and control beads (Flag peptide-coupled beads) were used to pull down recombinant TopBP1-0-5-WT or TopBP1-ΔBRCT4/5 (compare a(ii)). SDS gels were silver stained. kDa specifies molecular weight marker bands. The experiment was done more than three times with similar results. Source data are provided as a Source data file.

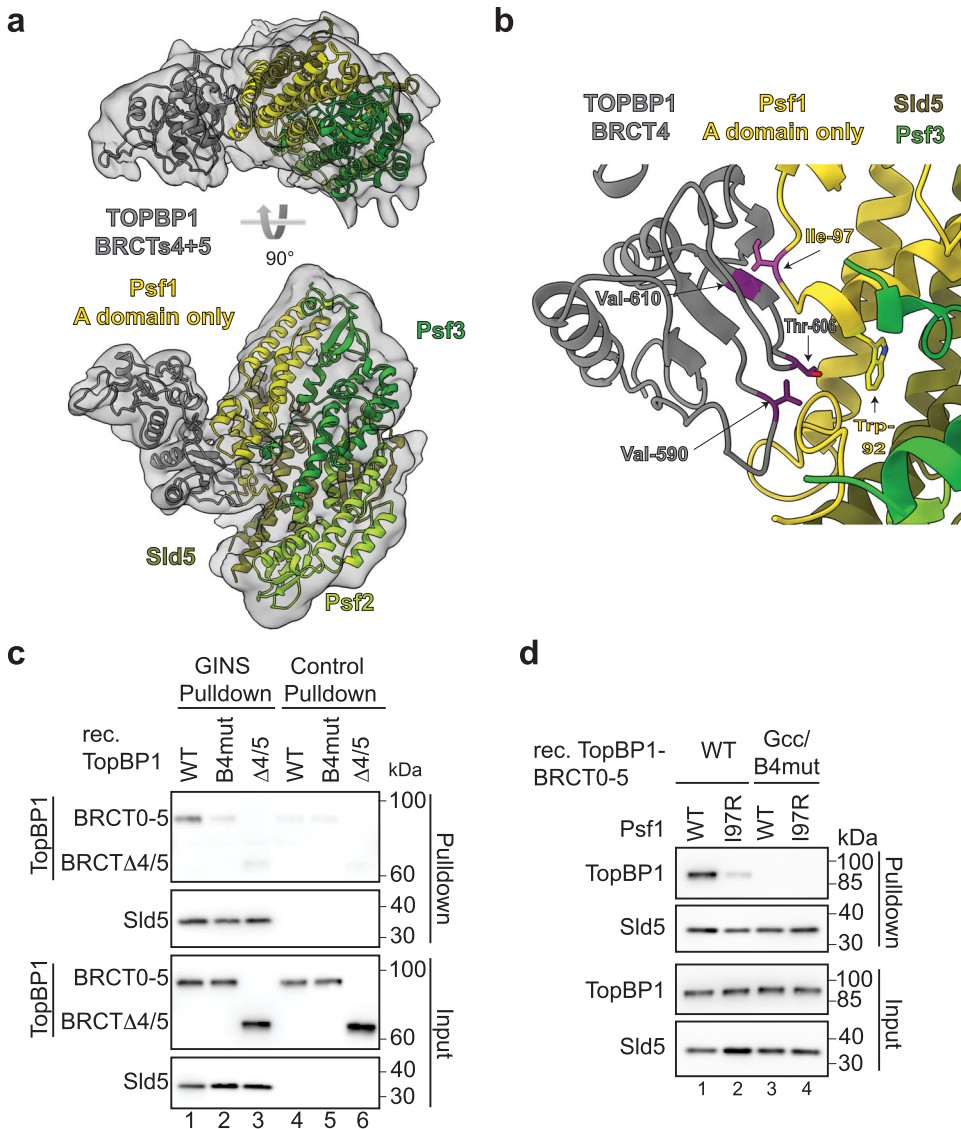

**Fig. 2 | Protein structure of the GINS·TopBP1-BRCT4/5 complex. a** Structural model of the GINS-TopBP1-BRCT4/5 complex. Crystal structures of the GINS subunits (PDB:2E9X) (shades of green, yellow), and the central BRCT4/5 domain of TopBP1 (PDB:3UEN) (grey) were docked into the cryo-EM volume shown as transparent volume. **b** Zoom-in of the BRCT4-Psf1 interface. The residues in stick representation appear crucial for the interface. Mutations to break the interaction are coloured pink (Psf1-I97R) and purple (TopBP1-B4mut; compare Fig. 1a(ii)). **c** Pulldown of the indicated recombinant TopBP1-BRCT0-5-strep versions (see

Fig. 1a(ii)) by Flag-beads-immobilised recombinant GINS or Flag peptide-coupled control beads. Analysis was done by immunoblotting. The experiment was done more than three times with similar results. **d** Pulldown of the indicated recombinant TopBP1-BRCT0-5-WT-strep using immobilised GINS-WT or GINS carrying a Ile97 to arginine mutation in Psf1.TopBP1-Gcc/B4mut-strep (Figs. 1a(ii) and 3a) was used as a non-GINS binding control. The experiment was done twice with similar results. Source data are provided as a Source data file.

crystal structure by rigid-body docking (PDB:2E9X) (Fig. 2a). As features of the individual GINS subunits were clearly identifiable, this also allowed unambiguous orientation of the pseudo-symmetric complex. The crystal structure appeared to account for all the density seen with the exception of a protrusion away from the main body of the GINS complex. This remaining density was too large to represent just the B domain of Psf1 (absent from the docked X-ray crystal structure), or the single BRCT3 of TopBP1, but was readily accounted for by docking of the X-ray crystal structure for TopBP1-BRCT4/5 (PDB:3UEN). The two BRCT domains were arranged such that BRCT4 interacted directly with the Psf1 subunit of GINS.

The GINS interacting region within the BRCT4 repeat of TopBP1 faces towards the A domain of Psf1 and the linker region connecting domains A and B (Fig. 2a, b), close to where the B domain docks to the CMG helicase (B domain not resolved). Note that some

ambiguity exists as to the precise molecular details due to the moderate resolution. In BRCT4, the interaction surface involves a region around beta strands β3 and β4. This region diverts from the canonical BRCT fold in that it lacks an alpha helix (α2)[44] as if to create space for GINS binding (Supplementary Fig. 5a, b). The region shows high sequence conservation in BRCT4 equivalents across a wide range of metazoans (Supplementary Fig. 5c). The GINS interaction surface in BRCT4 consists of residues found at the edge of the central four-stranded β-sheet, with additional interactions made by residues in the loops connecting the strands, including amino acids in the variant region between strands β3 and β4. Together, TopBP1-BRCT4 binds to the A domain of Psf1 and involves the linker region located towards the B domain. To our knowledge, this represents a novel mode for a BRCT domain-mediated interaction.

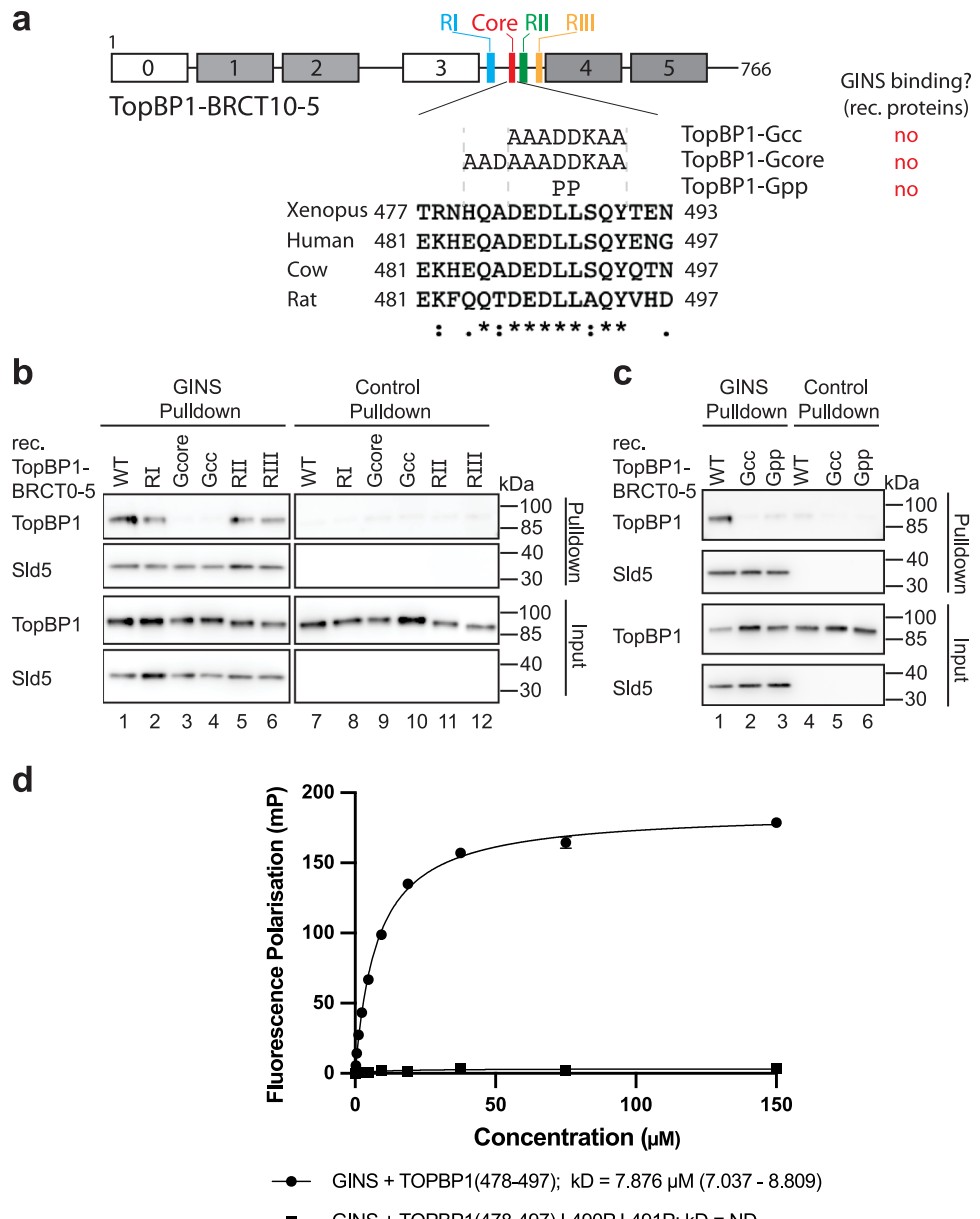

**Fig. 3 | The conserved core of the TopBP1-GINI domain is required for a stable interaction with GINS. a** Domain model and T-COFFEE alignment (one-letter aa code) of TopBP1 proteins from humans (Q92547) *Xenopus laevis* (Q7ZZY3), *Bos taurus* (A0A3Q1LWE4) and *Rattus norvegicus* (A0A8I6GFZ6). Coloured boxes indicate GINI region mutants in sub-regions I (blue), II (green) and III (orange), and in the GINI core helix (red) (details in Supplementary Fig. 7). Amino acid substitutions, names of GINI core region mutations and corresponding GINS binding capabilities are shown. ., : and * indicate low conservation, high conservation and identical amino acids, respectively. **b/c** Pulldown of recombinant TopBP1-BRCT0-5-strep-WT or the indicated GINI region mutants (**a**) by bead-immobilised GINS or Flag peptide-coupled control beads. Analysis was done by immunoblotting. The experiments were done more than three times with similar results. Source data are provided as a Source data file. **d** Fluorescence polarisation measurements including dissociation constants (kD) calculations using recombinant GINS tetramer and fluorescently labelled TopBP1-GINI-core peptides (WT or Gpp (**a**)). For each peptide *n* = 3 independent experiments and data are presented as mean values +/−SEM.

TopBP1 residues Val590, Thr606 and Val610 appeared to be involved in the interface, providing a series of direct hydrophobic interactions with the side chains of Trp92 and Ile97 in Psf1 (Fig. 2b). We mutated the three TopBP1 residues to glutamate, arginine and glutamate, respectively, and tested if the resulting protein (TopBP1-B4mut; Fig. 1a(ii)) co-purified with GINS. Here, TopBP1-B4mut was strongly compromised in its ability to bind GINS (Fig. 2c) but could still interact with phospho-53BP1 (Supplementary Fig. 6a). We also altered the opposing Psf1 interface by mutating Ile97 to an arginine (Psf1-I97R). Recombinant Psf1-I97R bound TopBP1 weaker than the wild type protein (Fig. 2d), but could still form GINS tetramers (Supplementary Fig. 6b).

## TopBP1 requires the conserved core of the GINI region for GINS binding

As the predicted interface between the TopBP1-GINI region and GINS[32] was not evident in our cryo-EM map we used additional interaction experiments to validate a role for this motif in GINS binding. The GINI region had been loosely defined as sitting between TopBP1-BRCT3 and BRCT4 (Fig. 1a(i)). Multiple sequence alignment across this region revealed a high degree of conservation between vertebrate species with the section from Glu484 to Tyr494 consisting of almost completely identical amino acids (Fig. 3a, Supplementary Fig. 7). We mutated this core, and other regions of conservation, to generate the mutants TopBP1-Gcc (GINI centre core), Gcore (GINI core) and RI-III

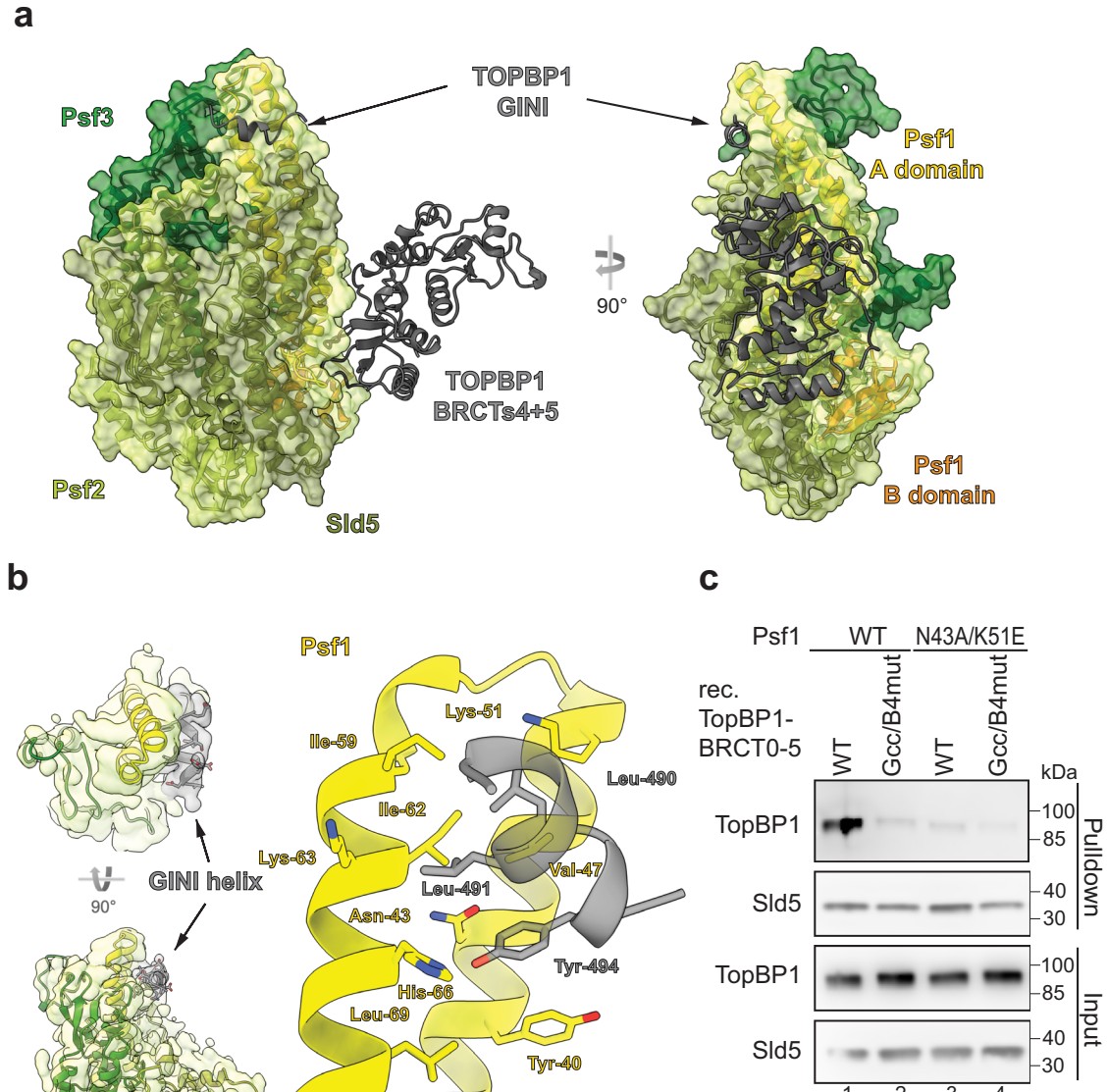

**Fig. 4 | The TopBP1-GINI domain interacts with Psf1-A domain. a** Structural model of TopBP1-BRCT3-5 complexed to the GINS tetramer predicted using AlphaFold2-Multimer. All chains are represented as cartoon. The transparent volume represents the surface of the GINS complex (not cryo-EM density). Of TopBP1, only the GINI helix (residues 485–496) and BRCT4/5 (residues 551-766), are shown. **b** Left: Cryo-EM volume showing extra helical density ascribed to the GINI helix (grey). Right: a structural model based on this cryo-EM and an AlphaFold2-Multimer predicted structure. **c** Pulldown of recombinant TopBP1-BRCT0-5-WT-strep using immobilised GINS-WT or GINS carrying the indicated Psf1 mutations. TopBP1-BRCT0-5-Gcc/B4mut-strep served as a non-GINS binding control. Analysis was done by immunoblotting. The experiment was done twice with similar results. Source data are provided as a Source data file.

(regions I-III) (Fig. 3a, Supplementary Fig. 7). Pull-down experiments revealed that TopBP1-Gcc and Gcore, but not RI-III, were compromised in GINS interaction (Fig. 3b). Secondary structure prediction and AlphaFold2 models of TopBP1 suggested the presence of a short helix within this core region for some vertebrate species. By mutation of the central leucine-leucine motif to two consecutive prolines to break the helix (TopBP1-Gpp; Fig. 3a) we could show that the interaction with GINS was strongly compromised, with no effect on binding to 53BP1 (Fig. 3c, Supplementary Fig. 6a). To test if this helix constituted a GINS-interaction surface, we used fluorescence polarisation (FP). A fluorescently labelled GINI-core peptide (Ala478-Gly497) bound the GINS tetramer with a dissociation constant of $K_d$ of ~8 μM (Fig. 3d). A peptide harbouring the PP mutation failed to bind (Fig. 3d). We therefore concluded that the GINI region is a GINS interaction site in its own right, but with moderate binding affinity.

## The GINI-core helix binds Psf1 and 3

We next sought to identify the site in GINS that interacts with the TopBP1-GINI region. Structural prediction using AlphaFold2-Multimer suggested that the GINI-core helix interacted with the distal part of the Psf1 A domain (Fig. 4a, Supplementary Fig. 8). Moreover, blind molecular docking simulations with the GINI core region (487-DEDLLSQY-494) resulted in the top-10 solutions clustering in the region of the AlphaFold-predicted GINI binding site (Supplementary Fig. 9).

A BS[3]-cross-linking mass spectrometry experiment with purified recombinant TopBP1-GINS also confirmed the predicted proximity of the TopBP1-GINI core helix region (residues 487–494) and the Psf1-A domain. Several GINI-core-proximal residues cross-linked with Psf1-Lys63 and Thr61 (Supplementary Fig. 10a, Supplementary Data 1); both cross-linked Psf1 residues are situated at the distal end of the Psf1 A domain (Supplementary Fig. 10b, c). The GINI-core-proximal residues

also cross-linked with lysines 74 and 80 of Psf3; each Psf3 residue also being in close proximity to the predicted site of interaction. Given that the GINI-core helix itself contains no lysine residues, our data are consistent with the hypothesis that this helix binds to both the distal region of the Psf1 A domain and the adjacent region of Psf3 (Supplementary Fig. 10 provides a detailed description).

With all this information in hand, we could improve the resolution of our cryo-EM data, using an alternative GINS-expression construct, which omits the Psf1-B domain, and served to improve detail around the predicted interaction region (Supplementary Fig. 11). The resultant map showed a volume of extra density that allowed a short helical element to be incorporated into our model, consistent with the AlphaFold2-Multimer predictions (Fig. 4b). The hydrophobic side of the amphipathic GINI-core helix can be seen to pack across two helices of the Psf1-A domain, making interactions with the side chains of Leu490 and Leu491 of the GINI region and Val47, Lys51, Ile59, Ile62, Lys63 and His66 of Psf1. In addition, the side chain of Tyr494 in the GINI region is inserted into a small pocket on the surface of Psf1 formed by Tyr40, Asn43 and Leu69 with the hydroxyl group picking up a hydrogen bond with the side chain of His66.

For validation, we mutated Asn43 and Lys51, to Ala and Glu, respectively. Asn43 forms part of the pocket accepting Tyr494 from the GINI region, whilst Lys51 is positioned such that it could form a salt bridge with either Asp487 or Glu495 and part of the hydrophobic grove that accommodates the leucines of the GINI-core helix. Psf1-N43A/K51E readily formed the GINS tetramer (Supplementary Fig. 6b) but was severely compromised in its ability to interact with TopBP1-BRCT0-5-WT (Fig. 4c), again consistent with binding of the GINI region to the distal end of the Psf1-A domain and the adjacent regions in Psf3.

## Simultaneous inactivation of the GINI core region and BRCT4 is required to suppress replication origin firing

Our pulldown interaction studies with GINS and TopBP1-BRCT0-5 carried out at 150 mM sodium chloride suggested that the GINS interacting regions GINI and BRCT4 of TopBP1 are individually required for GINS interaction (Figs. 2c and 3c). However, pulldowns in sodium acetate revealed specific interactions of the individual interaction regions, albeit at reduced levels compared to TopBP1-WT (Fig. 5a, Supplementary Fig. 12a). Here, the GINI region bound to GINS more efficiently than BRCT4. Estimation of binding free energies suggested that the affinities are lower when the surfaces interact individually rather than simultaneously (Fig. 5b). Together, this suggests a model in which both interactions have relatively moderate binding affinities, but cooperate to form a composite interaction surface with high avidity.

We next asked how the TopBP1-GINI and BRCT4 regions contribute to genome replication, by testing if mutational inactivation of GINI and BRCT4 (Fig. 5c) could suppress replication in *Xenopus* egg extracts. Immunodepleting TopBP1 with two independent antibodies (anti-TopBP1-#1 and #2) effectively removed the endogenous TopBP1 (Fig. 5d(i)), but neither GINS, MTBP, DNA Polε nor Cdc45 (Fig. 5d(ii)), from the extracts. TopBP1 depletion strongly reduced incorporation of radioactive $\alpha$-$^{32}$P-dCTP when compared to a mock IgG depletion control (Fig. 5e). Adding recombinant TopBP1-BRCT0-5-WT effectively rescued replication in depleted extracts (Fig. 5e), whereas addition of either TopBP1-BRCT0-5-Gcc or TopBP1-BRCT0-5-Gpp (BRCT4 intact) led to moderately reduced nucleotide incorporation by 21% and 55% (120 min time point), respectively (Fig. 5f, Supplementary Fig. 13a), consistent with the observed reduced affinity of BRCT4 for GINS in the absence of a functional GINI region (Fig. 5a, lanes 1–3). TopBP1-ΔBRCT4/5 and TopBP1-B4mut (GINI domain intact) were not detectably compromised in their abilities to support replication (Fig. 5f; Supplementary Fig. 13b), as observed before[31,32], consistent with the GINI region binding GINS stronger than BRCT4 (Fig. 5a, lanes 2, 4; Supplementary Fig. 12a). Combination of BRCT4/5 deletion or

mutation (B4mut) with either TopBP1-BRCT0-5-Gcc or -Gpp decreased replication to nearly background levels (Fig. 5f, Supplementary Fig. 13), consistent with these double mutants eliminating GINS binding (Fig. 5a, lanes 5, 6; Supplementary Fig. 12a). These results indicate that both TopBP1 binding regions, GINI and BRCT4, are individually capable of supporting replication, whilst simultaneous inactivation of the GINS binding regions strongly suppresses replication.

We next tested if the TopBP1-GINS interaction is required for replication origin firing. We isolated chromatin from *Xenopus* egg extracts upon TopBP1 immunodepletion and add-back of TopBP1-BRCT0-5-WT or mutants, 75 min after starting replication (a time point when genome replication has progressed significantly). To increase signals, we added aphidicolin, which prevents termination, allowing replisomes to accumulate on chromatin. Mass spectrometry using the CHROMASS protocol[45] and immunoblotting showed chromatin binding of Mcm2-7 subunits in TopBP1-depleted egg extract (Fig. 6a, c, d, Supplementary Fig. 14, Supplementary Data 2). Addition of geminin, an inhibitor of origin licensing[46], prevented Mcm2-7 binding (Fig. 6a, c, d). Together, this confirmed that origin licensing is independent of TopBP1. Several replisome markers, including GINS and Cdc45, were detected on chromatin in TopBP1-depleted extracts containing TopBP1-WT, but were significantly reduced when the TopBP1-Gcc-ΔBRCT4/5 mutant or buffer were added (Fig. 6a, b, d). Thus, the GINS binding sites in TopBP1 are required for origin firing (replisome formation), more precisely for a step between MCM-DH formation and CMG formation. Chromatin samples upon complementation with the single mutant TopBP1-Gcc increased the levels of individual replisome markers weakly but detectably over geminin controls (PolE2 in Fig. 6d and Cdc45 in Supplementary Fig. 12b), The TopBP1-ΔBRCT4/5 mutant supported strong replisome formation with only mild effect on Sld5, PCNA, PolE2 and Cdc45 compared to WT (Fig. 6d, lanes 2, 5; Supplementary Fig. 12b). Together, this analysis showed that both binding sites are individually capable of supporting origin firing. They do so to different degrees, consistent with the GINI domain mediating stronger GINS interaction than BRCT4. Together, we conclude that both GINS binding sites in TopBP1 act in concert to assemble the CMG helicase. We noticed a surprisingly strong origin firing defect with the TopBP1-Gcc mutant, given that it supports replication to a significant level. It is feasible that compensatory fork acceleration partially compensates for decreased replisome numbers in the Gcc mutant. Our observation that deleting BRCT4 has only minor effects on replisome formation is also consistent with the scenario that the BRCT4-GINS interaction has a role downstream of origin firing that aggravates the DNA synthesis defects (Fig. 5f, Fig. S13) of GINI site mutants.

## Mutually exclusive binding of TopBP1 and DNA Polε to GINS in the context of CMG

We next explored the relationship between the interactions of TopBP1 and DNA Polε with GINS. These proteins are part of the recently characterised metazoan pre-LC[37–40]. Analysis of pre-LC suggests that GINS may simultaneously bind DNA Polε and TopBP1. When GINS is complexed with DONSON in the pre-LC context, binding sites in Psf1 for the TopBP1-GINI and BRCT4 regions are accessible, as suggested by published Alphafold multimer models[37]. These pre-LC models are less informative on how DNA Polε affects the TopBP1-GINS interaction. The N-terminal region of PolE2 (PolE2-N; aa 1–75) binds GINS mainly at the Psf1-B domain, with minor interactions with residues in the A-domain that partly overlap with the TopBP1-BRCT4 interaction surface (Fig. 7a–c). To test if the Psf1-B domain is required for TopBP1 interaction, we did pulldowns with GINS lacking the Psf1-B domain (Psf1-ΔB). These indicated that the Psf1-B domain is dispensable for interaction with TopBP1-WT (Supplementary Fig. 15a(i)). This could be due to a residual affinity of BRCT4 for Psf1-ΔB or due to the GINI-Psf1 interaction. We therefore analysed specifically the interaction of GINS with BRCT4 using the TopBP1-Gcc mutant. TopBP1-Gcc bound to GINS

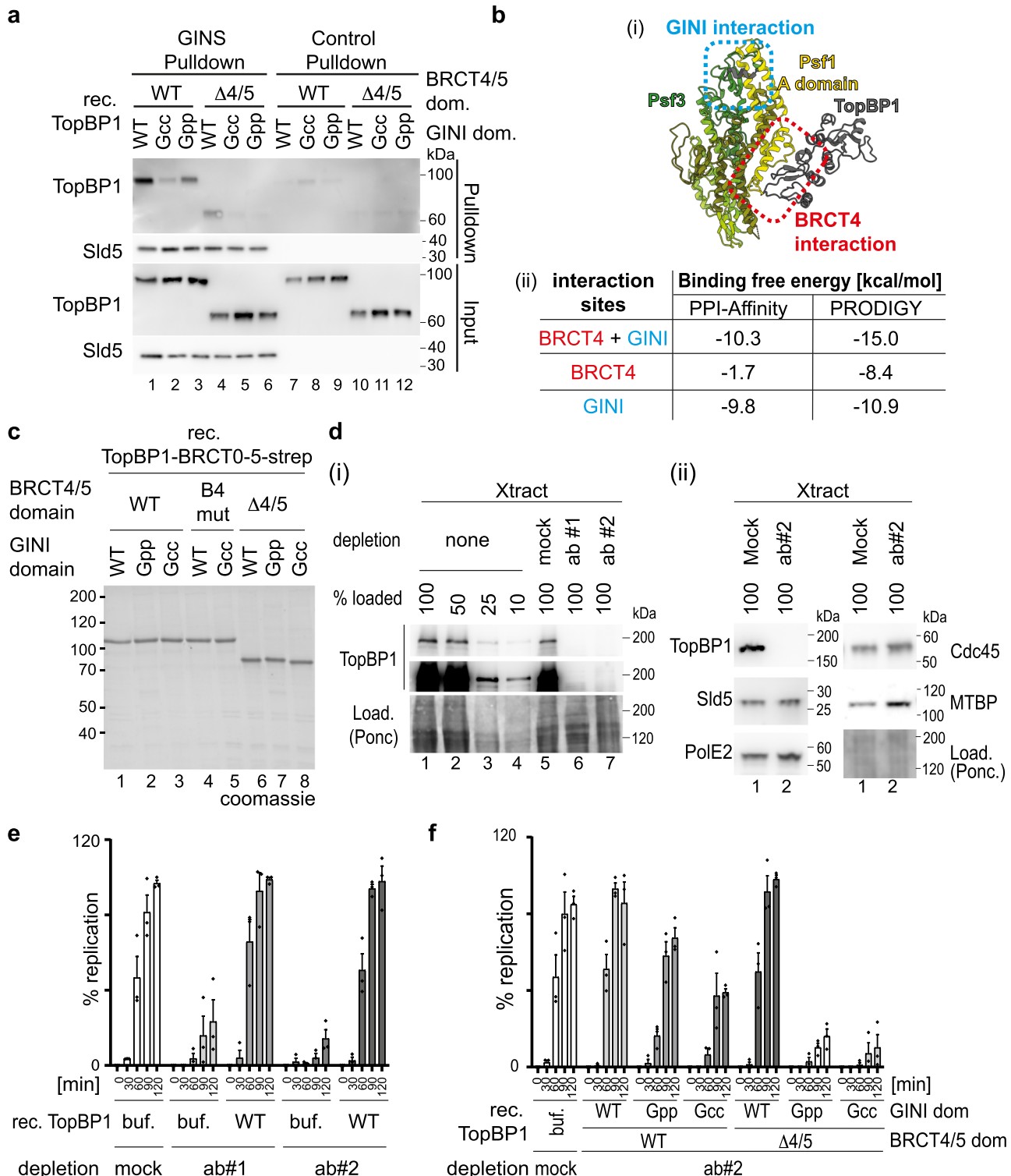

**Fig. 5 | Both GINS binding surfaces in TopBP1 cooperate in supporting genome replication. a** Immobilized recombinant Flag-GINS or Flag peptide-coupled beads were used to pulldown the indicated recombinant TopBP1-BRCT0-5-strep versions (Figs. 1a and 3a) using 100 mM NaAc in the binding buffer. The pulldown was analysed by immunoblotting. The experiment was done twice with similar results. **b** (i) shows the AlphaFold2-Multimer model of the GINS-TopBP1 complex that was used to estimate the binding free energy using PPI-Affinity and PRODIGY shown in (ii). Estimations were done for the binding interfaces involving TopBP1-GINI (blue) and TopBP1-BRCT4 (red), both together and alone. **c** Coomassie-stained SDS PAGE gel of the indicated recombinant TopBP1-BRCT0-5 proteins (Figs. 1a(ii) and 3a). **d** The indicated relative amounts (100%–10%) of immunodepleted and non-

depleted *Xenopus* egg extracts (Xtract) were analysed by immunoblotting using anti-TopBP1 #2 and other indicated antibodies. TopBP1 antibodies #1 and #2, or unspecific IgG (mock) were used for immunodepletions. Ponc, ponceau staining. Error bars, SEM, *n* = 3. The experiments were done more than three times (i) or twice (ii) with similar results. **e**, **f** Replication analyses (radioactive nucleotide incorporation) using *Xenopus* egg extracts immunodepleted with antibodies anti-TopBP1#1, #2 or unspecific IgG (mock). Buffer (buf), recombinant wild-type (WT) or mutant TopBP1-BRCT0-5-strep (**c**) were added. Diamonds, Individual data points of *n* = 3 independent experiments, Error bars, SEM. Source data are provided as a Source data file.

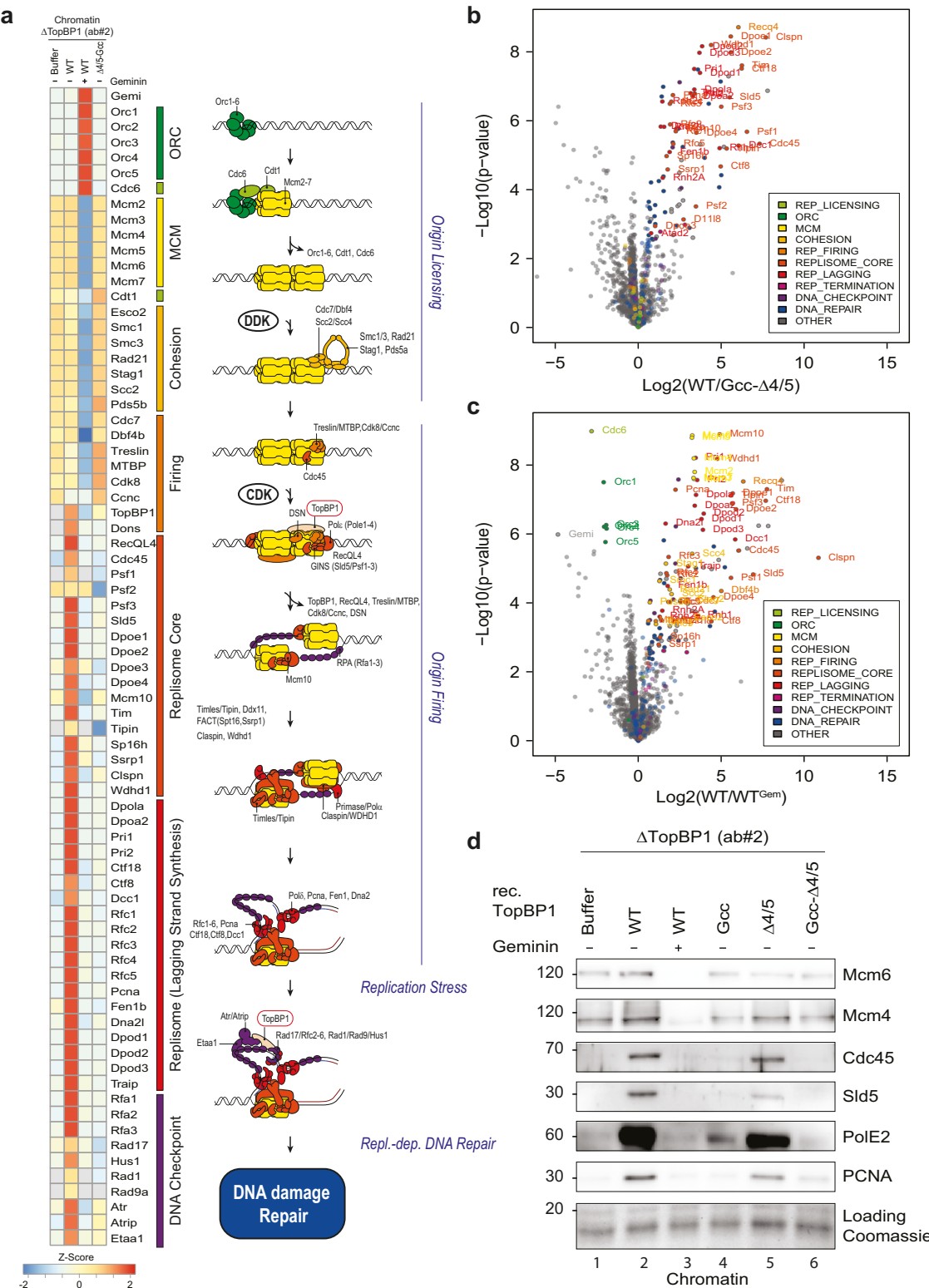

**Fig. 6 | GINS binding to TopBP1 is required for the origin firing step of replication. a–c** Heat map showing the intensity of chromatin-bound proteins recovered from TopBP1-depleted interphase *Xenopus* egg extracts supplemented wild type or mutant TopBP1 determined by quantitative mass spectrometry. Chromatin was isolated 75 min after sperm addition. All extracts contained 50 mg/ml aphidicolin and, where indicated, 2.25 mM geminin. The heat map shows log2 transformed label-free quantification (LFQ) intensities z-score normalized across rows and mean averaged over four independent replicates (*n* = 4). For the intensities of all proteins see Supplementary Data 2. **b, c** Volcano plots highlighting a selection of differentially detected proteins in samples described in (**a**). Log2 transformed LFQ intensities from four replicates each were analysed by a two-sided Student's t-test. Significantly changed proteins with an Benjamini–Hochberg adjusted *p*-value of equal or less than 0.05 are reported in Supplementary Data 2. For clarity, only proteins involved in DNA replication are labelled (all significantly enriched proteins are labelled in Supplementary Fig. 14). **d** Experiment as in (**a**) was analysed by chromatin isolation and immunoblotting with the indicated antibodies. Coomassie staining of the gel part containing histones served as a loading control.

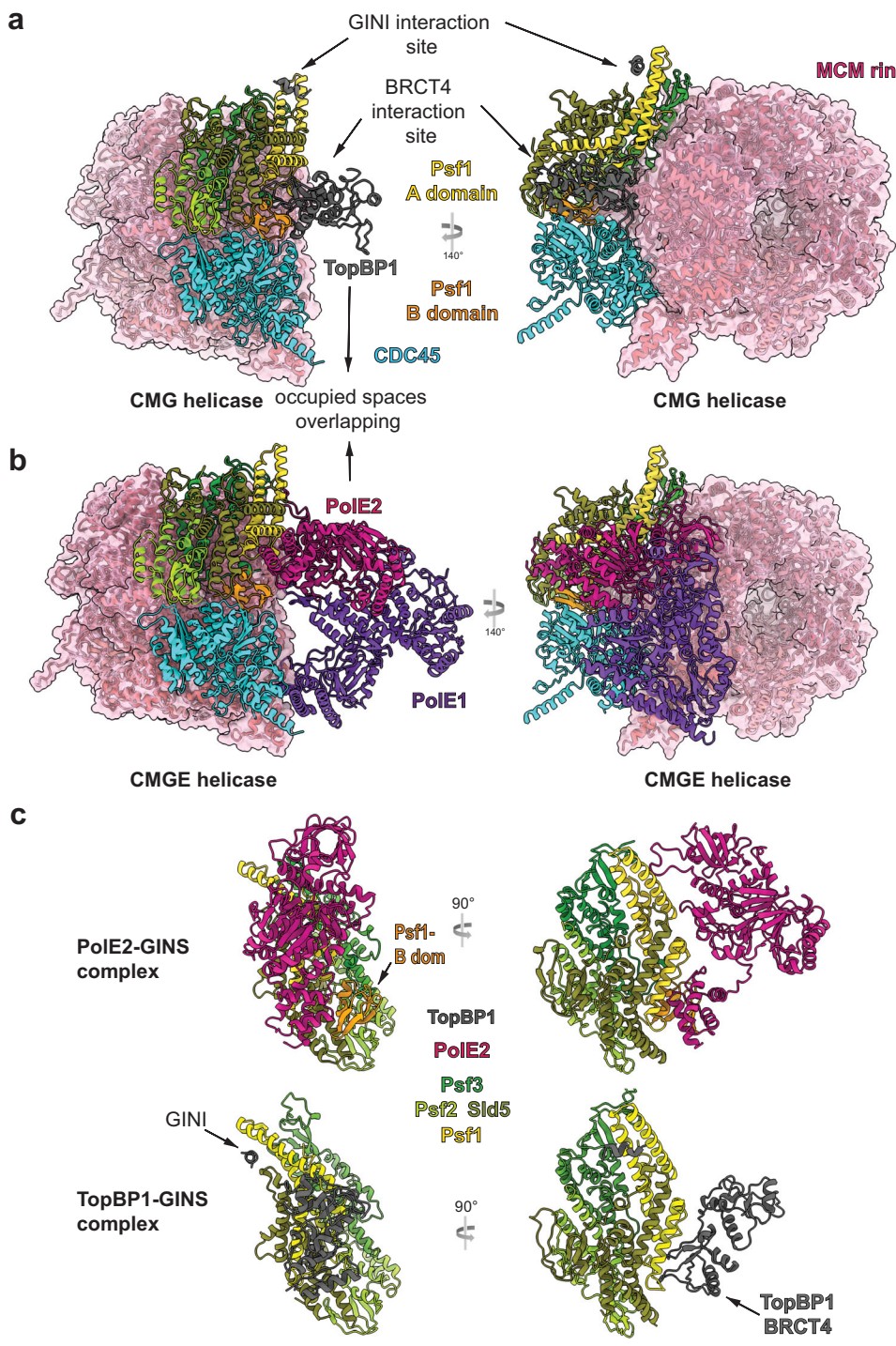

**Fig. 7 | Exclusive binding of TopBP1 and DNA polymerase epsilon to GINS in the CMG context. a**, **b** Structure of human CMG (PDB 7PFO), in panel (**a**) with our GINS-TopBP1 superposed, in panel (**b**) with DNA Polε (CMGE; PDB 7PFO) both shown from the same angle, demonstrating accessibility of the TopBP1 binding sites in CMG, and strong overlap between BRCT4 and PolE2 occupied spaces. The pink volume constitutes surface representation. **c** In-detail views of GINS-TopBP1 interaction sites from the cryo-EM model in this manuscript (8OK2) alongside the GINS-PolE2 interaction sites seen in the larger CMGE structure (7PFO).

partially dependently on the Psf1-B domain, whereas the B domain had no influence on TopBP1-WT and B4mut binding (Supplementary Fig. 15a(ii)). The moderate effect of the B domain deletion on BRCT4 binding is consistent with the partial overlap of the BRCT4 and PolE2 binding sites in Psf1. Consistently, TopBP1-Gcc, but neither TopBP1-WT nor B4mut, was partially competed off of immobilised GINS by ten times molar excess of PolE2-N (Supplementary Fig. 15b(ii)). TopBP1-WT was largely resistant to PolE2 competition also in stringent pulldown conditions (Supplementary Fig. 15b(i)). Together, in pre-LC, TopBP1 likely binds GINS through the GINI region, potentially also involving a residual affinity of BRCT4 for GINS despite the presence of DNA Polε. TopBP1 and GINS are likely further stabilised in pre-LCs via their interactions with DONSON[37,38,41].

Next, we analysed the CMGE helicase. Both TopBP1 binding sites in GINS are accessible in the CMG helicase, showing that TopBP1 could bind in the CMG context (Fig. 7a, b)[27]. However, the binding of DNA

Polε and TopBP1 seem mutually exclusive. Superposition of CMGE and CMG-TopBP1 revealed potential major steric clashes between TopBP1 and PolE2 around the BRCT4 interaction region (Fig. 7a–c). Based on this steric exclusion and the observed direct competition between PolE2 and BRCT4, we propose that TopBP1-BRCT4 disengages before DNA Polε associates to form CMGE. Because both TopBP1-GINS binding surfaces must cooperate for a strong GINS interaction BRCT4 disengagement should significantly weaken the TopBP1-GINS complex. Subsequent full dissociation of TopBP1 from CMGE may additionally require mechanisms that dislodge DONSON and Treslin-MTBP from CMGE.

The transition of GINS from TopBP1-bound to a stable CMG requires reconfiguration of the Psf1-B domain[47]. In CMG, the Psf1-B domain adopts a discrete conformation tightly packing against Cdc45 (Fig. 7a)[27]. In contrast, the B domain appears flexible in the GINS tetramer and in the TopBP1-GINS complex, because our cryo-EM data did not detect the B domain at the same map contour level as the rest of GINS (Supplementary Fig. 16a, b). A previous crystallization study of the GINS tetramer also suggested Psf1-B domain flexibility[48]. Thus, integration of GINS in the CMG appears to order the B domain. In this conformation, the B domain binds to the N-terminus of PolE2 once DNA Polε associates to form CMGE[27].

Together, we envision that the Psf1-B domain reconfigures into a well-ordered state upon CMG assembly. TopBP1-BRCT4 must then dissociate to exchange with DNA Polε for CMGE formation, which likely helps TopBP1 dissociate. Once CMG has formed, DNA Polε seems to dynamically associate with CMG[25]. These observations suggest that the three processes (1) handover of GINS from TopBP1 to MCM-DHs, (2) DNA Polε association and (3) TopBP1 dissociation may be mechanistically intertwined.

## Discussion

Our data provide an understanding of the interaction between TopBP1[Dpb11] and GINS at the molecular level, and how this supports replication origin firing. In the past decade, insight into origin firing has largely come from studies focussed on understanding how the yeast Mcm2-7 helicase operates as a molecular machine to build replication forks[6,10–12,14,16,17,19,49–52]. Most recently, the aspect of how GINS and Cdc45 are loaded to activate the helicase through formation of the CMG complex has been addressed in more detail by describing the metazoan pre-LC[37–41]. Our studies here reveal an important aspect of pre-LC, the essential role of the TopBP1[Dpb11]-GINS interaction (Supplementary Fig. 17).

Formation of the CMG complex serves as a key regulatory step of origin firing. In metazoa and yeast, CMG formation involves the factors Treslin/Sld3, MTBP/Sld7, RecQL4/DONSON/Sld2, TopBP1/Dpb11 and the Mcm2-7 helicase itself. Apart from coupling origin firing to the S phase via S-CDK and DDK, these signal integrators are targeted by the DNA damage checkpoint and other regulatory pathways to mediate DNA damage-dependent origin firing control, execution of the temporal replication programme and fine-tuning of origin firing[53–67]. Such regulation helps avoid replication stress that is key to structural and numerical chromosomal instability in cancer cells[68].

TopBP1[Dpb11] is a scaffold protein. To mediate S phase coupling of origin firing, the N-terminal triple BRCT0-2 module of TopBP1[Dpb11] binds Treslin[Sld3], in a strictly phospho-dependant manner, via two CDK sites that are also conserved in budding yeast Sld3[Treslin21,22]. Despite this level of conservation, some aspects of origin firing in vertebrates are likely to have changed mechanistically when compared to the yeast paradigm: (1) DDK strengthens the interaction between TopBP1[Dpb11] and the Treslin-MTBP[Sld3-Sld7] complex, albeit via an unclarified mechanism[69], whereas a similar role for DDK in yeast has not been reported. (2) an N-terminal fragment that excludes the BRCT4/5 module of TopBP1[Dpb11] effectively supports replication in *Xenopus* egg extracts[31,32], whereas the equivalent BRCT3/4 module of Dpb11[TopBP1] is

essential, through CDK-mediated binding to phospho-Sld2[RecQL4/DONSON23]. The metazoan-specific DONSON protein may replace some roles of Sld2[37–40]. These mechanistic differences exemplify evolutionary flexibility in S-phase coupling, supported by findings in the *C. elegans* and fission yeast model organisms where the contributions of Treslin[Sld3] and RecQL4[Sld2] equivalents have changed during evolution[70,71]. Such flexibility may be supported by the partially redundant roles of S phase kinases (S-CDK and DDK) and their substrates (Treslin[Sld3], RecQL4[Sld2], Mcm2-7).

TopBP1[Dpb11] supports CMG formation via an interaction with the GINS complex. In budding yeast, Dpb11[TopBP1] and GINS are recruited to MCM-DHs as part of a pre-LC comprising Dpb11[TopBP1], GINS, DNA Polε and Sld2[RecQL4/DONSON 72]. Dpb11[TopBP1] appears to dock the pre-LC onto Sld3[Treslin]-Sld7[MTBP]-Cdc45 via the CDK-mediated Dpb11[TopBP1]-Sld3[Treslin] interaction; DDK is required for initial binding of Sld3[Treslin]-Sld7[MTBP]-Cdc45 onto MCM-DHs[14,19]. In metazoans, RecQL4[Sld2] is not part of the stable protein complex defined as pre-LC, which instead contains DONSON. DONSON is a scaffold protein with a series of interaction sites for TopBP1[Dpb11], GINS, DNA Polε and Mcm3 (Supplementary Fig. 17a)[37–40]. Our described TopBP1[Dpb11]-GINS interaction probably occurs in the context of pre-LCs.

The GINI region of TopBP1[Dpb11] that had been broadly defined as the region between BRCT3 and BRCT4, bound to Psf1 and 3 in Y2H experiments[32]. We show here that a short alpha-helix found at the core of the conserved GINI region (GINI-core helix) forms an essential part of the binding surface that is necessary and sufficient for detectable interaction with GINS. This helix cooperates with a second binding site situated in BRCT4 to form a composite high-affinity GINS-interaction surface; involving a hereinto unknown mode of BRCT-interaction that relies on a specific type of BRCT fold found in TopBP1-BRCT4, which lacks helix 2. Notably, our in vitro experiments show that each individual GINS interaction site is capable of binding GINS, albeit with reduced efficiency compared to the complete composite interaction surface. This was consistent with the two GINS binding sites being able to support significant levels of replication in *Xenopus* egg extracts, which is in line with an earlier study focussing on the GINI region[32]. These results indicate that even a low affinity TopBP1[Dpb11]-GINS interaction is sufficient to allow origin firing. In vivo, contacts between other pre-LC components may compensate for a weak TopBP1[Dpb11]-GINS interaction. A high degree of inter-dependence between association of the components was also observed for the yeast pre-LC complex[72]. Simultaneous mutation of both GINS binding domains in TopBP1[Dpb11] suppressed origin firing. This suggests that neither interaction site has a unique molecular role beyond GINS binding. The simplest scenario is that their common essential molecular role is to bind GINS chaperoning it onto MCM-DHs.

An alternative interpretation of our TopBP1[Dpb11] mutant analysis seems feasible: TopBP1-Δ4/5 and B4mut only resulted in no DNA synthesis defect and a weak origin firing defect. This scenario is consistent with the possibility that the BRCT4-GINS interaction plays an unidentified role in replication, for example in elongation, which could contribute to the observed effect that BRCT4 inactivating mutants aggravate the defects of GINI site mutants in DNA replication assays (nucleotide incorporation).

Consistent with published Y2H experiments[32], our structural data indicate that the GINI core helix region interacts with both Psf1 and Psf3, at a position close to the distal end of the Psf1 A domain, whereas BRCT4 interacts with both the base of the Psf1-A domain and the linker that connects it to the B domain. Notably, the two receiving interfaces in GINS are fully accessible both in the free GINS tetramer and in the CMG context, consistent with a model where TopBP1[Dpb11] recruits GINS using both motifs (Supplementary Fig. 17b). TopBP1[Dpb11] may simultaneously bind phosphorylated Treslin[Sld3]-MTBP[Sld7] via its N-terminal BRCT0-2 module[21,22,73], with Treslin[Sld3] bringing Cdc45 along through its Sld3-Treslin homology domain[18,74]. One open question in the field

had been the biological role of the BRCT3 domain in TopBP1, which has no equivalent in Dpb11. We now know that an essential role of BRCT3 is the binding to DONSON[37,38].

After delivery of GINS and Cdc45 to form CMG, the subsequent release of pre-LC components including and of Treslin[Sld3]-MTBP[Sld7] is likely to be important. Whether DNA Polε is transferred from pre-LC to CMG to form CMGE during pre-LC dissociation is an interesting aspect. Release may help recycle these limiting factors to allow their use to fire late or dormant origins[35]. Moreover, their release from CMG may be essential for converting CMGs into replisomes. If release is essential, these firing factors constitute a firing-inhibitory intermediate as long as they are CMG-bound, and their controlled dissociation could be part of regulating origin firing. Dissociation could be triggered by the Mcm10 and RecQL4 firing factors that have roles downstream of CMG formation, or by CMG binding replisome factors like DNA polymerase alpha.

Dissociation and disassembly of Treslin[Sld3]-MTBP[Sld7]-TopBP1[Dpb11] may involve the action of phosphatases to counteract cell cycle kinase phosphorylation[36,69,75]. Moreover, the low affinities of the two individual GINS-interacting sites in TopBP1[Dpb11] offer the idea that release of one site would significantly destabilise the TopBP1-GINS complex to trigger TopBP1[Dpb11] dissociation. Competition of DNA Polε with TopBP1 binding may also be involved in TopBP1[Dpb11] dissociation, because simultaneous binding of the two proteins to CMG is sterically impossible without significant conformational reconfigurations. Whether TopBP1[Dpb11] dissociation is a prerequisite for DNA Polε association, or whether the polymerase acts to displace TopBP1[Dpb11] remains unclear. The reconfiguration of the Psf1-B domain during CMG formation suggests another potential mechanism for TopBP1[Dpb11] release. The B domain seems flexible in the TopBP1[Dpb11]-GINS complex and establishes an extensive set of interactions with Cdc45 when it reconfigures to form CMG, which might favour Polε binding over TopBP1[Dpb11]-BRCT4.

Recent insights have suggested that formation of the CMG complex is tightly associated with (i) splitting of Mcm2-7 double hexamers into single hexamers; (ii) the exchange of ADP for ATP in the Mcm2-7 nucleotide-binding sites; and (iii) limited untwisting of dsDNA inside the central channels[6,11,12]. How TopBP1[Dpb11], Treslin[Sld3] and MTBP[Sld7] relates to these steps will be key to understanding whether their mechanistic roles go beyond just a simple chaperoning function, guiding Cdc45 and GINS to MCM-DHs. For example, the reported homo-dimerization of TopBP1[Dpb11] and Treslin[Sld3]-MTBP[Sld7] and DONSON might play a role in the cooperative conversion of Mcm2-7 hexamers into replisomes, thus guaranteeing bi-directional replication[41,69,76].

Taken together, our data provide insight into, and a testable hypothesis for, the series of molecular events leading to CMG formation, and its regulation; potentially leading to future therapeutic routes for the treatment of cancer, due the high reliance of tumours on high levels of replication initiation[77]. Targeted interference with origin firing could be one potential avenue, consistent with the testing of DDK inhibitors as candidates in clinical trials[78].

## Methods
Uncropped images of SDS PAGE gels are available as a Source data file. Information on antibodies and plasmids used are available in Supplementary Data 3.

### Protein purification
**Purification of recombinant TopBP1.** Purification of TopBP1-331-766 (BRCT3-5) for structural studies was done using bacterial expression. BL21(DE3) cells carrying a pET17 plasmid encoding HIS-(3C)-TopBP1-331-766-Strep was were grown in TurboBroth at 37 °C to an OD600 of 2.0 and induced by with 0.5 mM IPTG prior to overnight incubation at 16 °C. Lysis by sonication was done in lysis buffer (25 mM HEPES pH 7.5, 200 mM NaCl, 0.5 mM TCEP, 10 U DNASE Turbo and complete, EDTA-

free Protease Inhibitor Cocktail (Merck). The resulting lysate was clarified by centrifugation at $40,000 \times g$ for 60 min at 4 °C. The supernatant was applied to a 5 ml HiTrap TALON crude column (GE Healthcare, Little Chalfont, UK), washed with lysis buffer, followed by a wash with lysis buffer containing 10 mM imidazole. Elution followed by increasing imidazole to 250 mM. The eluted protein was separated on a 5 ml HiTrap STREP column (GE Healthcare, Little Chalfont, UK), and eluted with 2 mM desthiobiotin. The eluate was concentrated before separation on a Superdex200increase 10 300 column (GE Healthcare, Little Chalfont, UK) equilibrated in 10 mM HEPES pH 7.5, 150 mM NaCl, 0.5 mM TCEP.

Recombinant TopBP1-1-766-Strep-WT (BRCT0-5) and mutants for interaction studies and experiments in *Xenopus* egg extract were purified using SF9 insect cells. The cells were grown in suspension in insect cell media (Pan biotech, P04-850 500) at 27 °C. Baculoviruses were generated using pLIB-based constructs (Addgene, 80610). 500 ml Sf9 cells ($1 \times 10^6$/ml) were infected with a 1:100 dilution of recombinant baculovirus carrying TopBP1-1-766-Strep-WT or mutants, and were incubated at 27 °C for 72 h. Cell pellets were lysed by douncing in 30 ml of lysis buffer (20 mM HEPES pH 8.0, 150 mM NaCl, 0.1% (v/v) Tween-20, 0.5 mM TCEP, protease inhibitor cocktail (Roche Complete protease inhibitor cocktail, 05056489001)). The lysate was centrifuged at $44,800 \times g$ for 45 min and the supernatant was loaded onto the StrepTrapHP-1ml column (Cytiva, 28907546). Elution was done using 2.5 mM desthiobiotin in elution buffer (20 mM HEPES pH 8.0, 150 mM NaCl, 0.01% Tween-20 (v/v), 0.5 mM TCEP, 2% (v/v) glycerol, 2.5 mM desthiobiotin).

**Purification of GINS.** GINS for structural and fluorescence polarisation studies was purified in SF9 insect cells Psf2, Psf3 and HIS-(3C)-Sld5 were expressed together with Psf1 or Psf1 (1–151; B domain deleted)) from a single baculovirus, respectively, produced using pBig1a. Cell pellets were resuspended in lysis buffer containing 25 mM HEPES pH 7.5, 200 mM NaCl, 0.5 mM TCEP, 10 U DNASE Turbo and complete, EDTA-free Protease Inhibitor Cocktail (Merck), then disrupted by sonication. The resulting lysate was clarified by centrifugation at $40,000 \times g$ for 60 min at 4 °C. The supernatant was separated on a 5 ml HiTrap TALON crude column (GE Healthcare, Little Chalfont, UK), using washing in lysis buffer supplemented with 10 mM imidazole. Elution was done using a gradient up to 250 mM imidazole. Peak fractions were pooled and concentrated. For subsequent size exclusion chromatography (analytical and for EM) a Superdex200increase 10 300 column (GE Healthcare, Little Chalfont, UK) equilibrated in 10 mM HEPES pH 7.5, 150 mM NaCl, 0.5 mM TCEP was used while for FP experiments a Superdex200 16 600 column (GE Healthcare, Little Chalfont, UK) equilibrated in 25 mM HEPES pH 7.5, 150 mM NaCl, 1 mM EDTA, 0.25 mM TCEP, 0.02% (v/v) Tween 20 was used.

GINS for pulldown experiments was purified using 6xHis-3xFlag-Sld5. Sf9 insect cells in suspension cultures were co-infected with 1:50 dilution of each four recombinant baculoviruses for expressing 6xHis-3xFlag-Sld5, Psf1, Psf2 and Psf3, respectively. The viruses were made using pLib-based plasmids. 72 h post infection, pellets from one-litre cells were lysed by douncing in 60 ml of lysis buffer (20 mM HEPES pH 8.0, 300 mM NaCl, 0.1% (v/v) Tween-20, 25 mM imidazole, 0.5 mM TCEP, protease inhibitor cocktail (Roche Complete protease inhibitor cocktail, 05056489001)). The lysate was centrifuged at $44,800 \times g$ for 45 min and the supernatant was incubated with 2.5 ml Ni-NTA agarose beads (Qiagen, 30210) for 1 h at 4 °C. The bound protein was eluted with six bead volumes elution buffer (20 mM HEPES pH 8.0, 300 mM NaCl, 0.01% (v/v) Tween-20, 250 mM imidazole, 0.5 mM TCEP, 2% (v/v) glycerol). The eluate was dialysed into 20 mM HEPES pH 8.0, 150 mM NaCl, 0.01% (v/v) Tween-20, 0.5 mM TCEP, 2% (v/v) glycerol, and then applied onto an HiTrap HP Q column (Cytiva, 17115301). To elute the bound protein, a linear gradient of NaCl (from 150 mM to 1 M) in elution buffer (20 mM HEPES pH 8.0, 0.01% (v/v) Tween-20, 0.5 mM

TCEP, 2% (v/v) glycerol) was applied. Peak fractions were pooled and separated by size exclusion chromatography (equilibration buffer: 20 mM HEPES pH 8.0, 300 mM NaCl, 0.01% (v/v) Tween-20, 0.5 mM TCEP, 2% (v/v) glycerol).

### Purification of PolE2-N and geminin from *E. coli*.

PolE2-N-1-75 (amino acids 1–75) with N-terminal MBP-TEV2 tag in pMal, and GST tagged *Xenopus* geminin (Xgeminin) in pGEX were expressed in Rosetta *E. coli* culture. Expressions were induced with 1 mM IPTG at OD600 = 0.6 at 20 °C (PolE2-N) or 25 °C (Xgeminin) overnight. Cells were harvested by centrifugation. For PolE2-N, cell pellets were resuspended in lysis buffer (20 mM Hepes pH 8.0, 150 mM NaCl, 0.5 mM TCEP, 2 (v/v) glycerol and protease inhibitor cocktail (Roche Complete protease inhibitor cocktail, 05056489001)) and lysed by sonication. The cell lysate was clarified by centrifugation. The supernatant was incubated with amylose resin (New England Biolabs, E8021S) for 1 h before elution with lysis buffer + 10 mM maltose. The peak fraction was dialysed into TopBP1 buffer (20 mM HEPES pH 8.0, 150 mM NaCl, 0.5 mM TCEP, 0.01% (v/v) Tween-20 and 2% (v/v) glycerol). For Xgeminin, lysis by sonication was carried out in 5 mg/ml lysozyme, 20 mM Hepes pH 7.7, 200 mM NaCl, 5 mM β-mercaptoethanol, 5% glycerin. After a clarifying centrifugation step, the supernatant was incubated with glutathione-sepharose (0.5 ml per liter culture; GE Healthcare 17513201) for 3 h before elution with lysis buffer + 50 mM glutathione (Applichem A2084,0025). Peak fractions were pooled and dialysed into XBE2 buffer (100 mM KCl, 2 mM MgCl$_2$, 0.1 mM CaCl$_2$, 1.71% w:v sucrose, 5 mM K-EGTA, 10 mM HEPES−KOH, pH 7.7) and concentrated to 360 μM Xgeminin. Aliquots were shock frozen and stored at −80 °C.

### Size exclusion chromatography for isolating the TopBP1-GINS complex

Recombinant TopBP1 and GINS proteins were mixed in an equimolar ratio to give a final concentration of 20 μM of each component and incubated on ice for at least 30 min prior to application to a Superdex200increase 10 300 column (GE Healthcare, Little Chalfont, UK) equilibrated in 10 mM HEPES pH 7.5, 150 mM NaCl, 0.5 mM TCEP.

### Fluorescence polarisation experiments

Fluorescein-labelled peptides (WT: Flu-GYGAPSEKHEQADEDLLSQ YENG or LLPP: Flu-GYGAPSEKHEQADEDPPSQYENG) (Peptide Protein Research Ltd, Bishops Waltham, UK) at a concentration of 100 nM, were incubated at room temperature with increasing concentrations of GINS in 25 mM HEPES pH 7.5, 150 mM NaCl, 1 mM EDTA, 0.25 mM TCEP, 0.02% (v/v) Tween 20 in a black 96-well polypropylene plate (VWR, Lutterworth, UK). Fluorescence polarisation was measured in a POLARstar Omega multimode microplate reader (BMG Labtech GmbH, Offenburg, Germany). Binding curves represent the mean of three independent experiments, with error bars representing SEM.

### Immunoprecipitations from transiently transfected 293T cells

Transient transfections of 6xMyc-Tev2-TopBP1 into 293T cells (ATCC CRL-11268) were carried out using PEI (polyethyleneimine) according to a protocol kindly shared by David Cortez' lab. 4 μg plasmid DNA in 100 μl DMEM (Thermofisher, 41965039) without penicillin-streptomycin were combined with 2.4 μl Polyethylenimin (Sigma, 408727; 10 mg/ml) and incubated for 20 min before addition to $4 \times 10^6$ cells on a 10-cm dish. Transfected cells were used 72 h post transfection. 72 h post transfection, 293T cells were lysed by douncing in ten times cell pellet volume of lysis buffer (20 mM HEPES pH 8.0, 150 mM NaCl, 0.1% (v/v) Tween-20, 0.5 mM TCEP, 2% (v/v) glycerol, protease inhibitor cocktail (Roche, Complete protease inhibitor cocktail, 05056489001)), and centrifuged at $20,000 \times g$ for 15 min at 4 °C. The soluble supernatant was added to magnetic anti-Myc beads (5 μl slurry per sample; Thermofischer, 88842) and incubated for 2 h at 4 °C rotating. Cells equivalent to 50% of a 15 cm dish were used per sample.

Beads were washed three times in lysis buffer with 5 min incubation each, and finally boiled in 50 μl Laemmli loading buffer (6.5% glycerol, 715 mM β-mercaptoethanol, 3% SDS, 62.5 mM Tris-HCl pH 7.9, 0.005% bromphenol blue).

### Pulldown experiments of recombinant PolE2-N or TopBP1 using immobilised GINS

Magnetic anti-Flag beads (1 μl slurry for western blot/silver staining-scale experiments, 4 μl for Coomassie-scale experiments) were coupled with 600 ng (western/silver-scale) or 2400 ng (coomassie-scale) GINS via 3xFlag-Sld5. 100 nM final concentration of TopBP1-BRCT0-5-Strep (WT or mutants) in 20 μl (western/silver-scale) or 80 μl (coomassie-scale) of interaction buffer (20 mM HEPES pH 8.0, 150 mM NaCl (or 100 mM NaCH$_3$COO when indicated), 0.01% (v/v) Tween-20, 0.5 mM TCEP, 2% (v/v) glycerol) were added to GINS-coupled beads in 5 μl (western/silver-scale) or 20 μl (coomassie-scale) interaction buffer. For PolE2-N, 1 μM PolE2-N protein in 20 μl were used for GINS-coupled beads in 5 μl reaction buffer. After incubation for 45 min at 4 °C rotating, beads were washed three times for 5 min with interaction buffer and boiled in 50 μl Laemmli buffer before SDS PAGE.

### Pulldown from cell lysates with recombinant TopBP1-1-766-strep (BRCT0-5)

For pulldown assays from soluble lysates of non-transfected 293T cells, 5 μl streptactin Sepharose HP (Cytiva, 28935599) beads were coupled with 10 μg recombinant TopBP1-1-766-strep-WT or mutants in coupling buffer (20 mM HEPES pH 8.0, 150 mM NaCl, 0.1% (v/v) Tween-20, 0.5 mM TCEP, 2% (v/v) glycerol) for 45 min at 4 °C before washing two times with coupling buffer and then once in cell lysis buffer (20 mM HEPES pH 8.0, 150 mM NaCl, 0.1% (v/v) Tween-20, 0.5 mM TCEP, 2% (v/v) glycerol, protease inhibitor cocktail (Roche Complete protease inhibitor cocktail, 05056489001). Control pulldowns were done with Flag-peptide-coupled beads or, when indicated, using GINS-coupled beads and the non-GINS binding TopBP1-Gcc-B4mut. 293T cells were lysed in ten times pellet volume of cell lysis buffer, and centrifuged at $20,000 \times g$ for 15 min at 4 °C. Soluble lysate from cells equivalent to 50% of a 15 cm tissue culture plate was added to the TopBP1-coupled beads and incubated for 2 h at 4 °C rotating. Beads were washed three times with 5 min incubation in lysis buffer and boiled in 50 μl Laemmli buffer before SDS PAGE.

### Generation of replicating *Xenopus* egg extracts

Our work with *Xenopus laevis* frogs complied with ethical regulations. The protocols used in this study, namely to handle the frogs, collect their eggs for cytosolic extract generation, and to prepare sperm, were approved by the Landesamt für Natur-, Umwelt- und Verbraucherschutz, Nordrhein-Westfalen (81-02.05.40.20.050).

*Xenopus laevis* egg extracts were prepared from metaphase II arrested eggs as described in ref. [79]. After washing with MMR (100 mM NaCl, 2 mM KCl, 1 mM MgCl$_2$, 2 mM CaCl$_2$, 0.1 mM EDTA, 5 mM HEPES−NaOH, pH 7.8) eggs were dejellied (2% cysteine w:v in H$_2$O, pH 7.8 with NaOH). Eggs were rinsed in XBE2 (100 mM KCl, 2 mM MgCl$_2$, 0.1 mM CaCl$_2$, 1.71% w:v sucrose, 5 mM K-EGTA, 10 mM HEPES−KOH, pH 7.7). Then, the eggs were transferred into centrifuge tubes containing 1 ml XBE2 + 10 μg/ml protease inhibitor (aprotinin Sigma A6279, leupeptin Merck 108975, pepstatin Carl Roth 2936.1) + 100 μg/ml cytochalasin D (Sigma C8273). For packing, eggs were centrifuged for 1 min at $1400 \times g$ at 16 °C in a swingout rotor. Excess buffer, activated and apoptotic eggs were removed before crushing the eggs at $16,000 \times g$ for 10 min at 16 °C. The extract was collected with a 20 G needle and supplemented with 10 μg/ml cytochalasin D, 10 μg/ml protease inhibitors, 1:80 dilution of energy regenerator (1M phosphocreatine K salt, 600 μg/ml creatine phosphokinase in 10 mM HEPES−KOH pH 7.6) and LFB1/50 (10% w-v sucrose, 50 mM KCl, 2 mM MgCl$_2$, 1 mM EGTA, 2 mM DTT, 20 mM K$_2$HPO$_4$/KH$_2$PO$_4$ pH 8.0, 40 mM

HEPES–KOH, pH 8.0) to 15% (v:v). The extract was cleared at $84,000 \times g$ at 4 °C for 20 min in a swingout rotor. The cytoplasmatic layer was collected and supplemented with 2% glycerol v:v. The extract was frozen by dropping 20 μl aliquots into liquid nitrogen and stored at −80 °C.

## Immunodepletion of TopBP1 from *Xenopus* egg extract

For XTopBP1 depletion from *Xenopus* egg extract, 0.5 μg antibody #1, #2 or IgG (rabbit; self-made) were coupled per μl magnetic Protein G dynabeads (Life Technologies 10004D) for 1 h at RT. Freshly thawed extracts were supplemented with 1/40 energy regenerator and 250 μg/ml cycloheximide and released into interphase by adding 0.3 mM $CaCl_2$ for 15 min at 23 °C. Subsequently, extracts were depleted for 45 min on ice with 0.675 μg antibody per μl extract. After depletion the extract was aliquoted, snap frozen and stored at −80 °C for replication assays and chromatin isolations.

## *Xenopus* egg extract replication assays

For TopBP1 addback experiments, 6 ng/μl recombinant TopBP1 protein or buffer were added to TopBP1 immunodepleted interphase *Xenopus* egg extract containing 50 nCi/μl $\alpha^{32}$P-dCTP (Perkin Elmer NEG513A250UC). To start the replication reaction the egg extracts were supplemented with 5 ng sperm DNA (kindly provided by the lab of O. Stemmann) per μl extract and incubated at 23 °C. 1 μl extract was spotted for each time point onto a glass fibre membrane. Unbound $\alpha^{32}$P-dCTP was removed by rinsing the membrane three times 15 min with ice cold 5% TCA in water and once with ice cold EtOH. Newly replicated DNA was detected by phospho-imaging. Standard error of the mean was calculated from three independent experiments.

## Chromatin isolation from *Xenopus* egg extracts and analysis by mass spectrometry (CHROMASS) and western blotting

For chromatin isolation, XTopBP1-depleted interphase egg extract was supplemented with 50 μg/ml aphidicolin and aliquoted into 15 μl samples before addition of buffer or recombinant TopBP1 to a final concentration 6 ng/μl. 2.25 μM Xgeminin or buffer were added before incubation for 10 min on ice. To start the replication, 9 ng sperm DNA/μl extract was added and incubated for 75 min at 23 °C. Reactions were stopped with 300 μl ice cold ELB salt (10 mM HEPES–KOH pH 7.7, 50 mM KCl, 2.5 mM $MgCl_2$) + 250 mM sucrose + 0.6% triton X-100. The diluted extract was loaded onto a sucrose cushion (150 μl ELB salt + 25% sucrose) and centrifuged for 10 min at $2500 \times g$ at 4 °C in a swingout rotor. The supernatant above the sucrose cushion was removed and the cushion surface was washed twice with 200 μl ELB salt containing 250 mM sucrose. The cushion was removed leaving about 15 μl behind, followed by centrifugation for 2 min at $10,000 \times g$ at 4 °C in a fixed angle rotor. For western blotting, the chromatin pellet was resuspended in 24 μl 1x Laemmli SDS sample buffer (6.5% glycerol, 715 mM β-mercaptoethanol, 3% SDS, 62.5 mM Tris-HCl pH 7.9, 0.005% bromphenol blue) and 5 μl were separated by denaturing SDS PAGE. For MS analysis, four biological replicates of chromatin pellets were analysed by CHROMASS[45]. In brief, chromatin pellets were resuspended in 50 μl denaturation buffer (8 M Urea; 100 mM Tris pH 8). Dithiothreitol (DTT) was added to a final concentration of 5 mM and samples were incubated at 22 °C for 30 min. To alkylate peptides, iodoacetamide (20 mM) was added and samples were incubated at 37 °C for 30 min. DTT (25 mM) was added, and samples were incubated at 22 °C for 5 min. 500 ng Lys-C (add Supplier) were added, and samples were incubated at 37 °C for 3 h. Samples were diluted with 100 mM ammonium bicarbonate to adjust the urea concentration to 1 M. Trypsin (Sigma, 500 ng per sample) was added and samples were incubated over night at 37 °C. Digested peptides were acidified with 1% trifluoroacetic acid (TFA) and desalted on Empore C18 material according to Rappsilber et al. https://www.nature.com/articles/nprot.2007.261). Eluted peptides were dried in a vacuum concentrator and reconstituted with 9 μl

of buffer A (0.1% formic acid) containing 0.2% acetonitrile and 0.01% trifluoro acetic acid. For MS analysis, 4 μl of the reconstituted peptides were separated on an EASY-nLC™ 1200 chromatography system (Thermo Scientific) coupled to a Q Exactive HF-X Orbitrap LC-MS/MS System (Thermo Fisher Scientific) via a Nanoflex ion source. Peptide separation was carried out in analytical columns (50 cm, 75 μm inner diameter packed in-house with C18 resin ReproSilPur 120, 1.9 μm diameter Dr. Maisch) using a 3-h nonlinear gradient at a flow rate of 250 nl/min using buffer A (aqueous 0.1% formic acid) and buffer B (80% acetonitrile, 0.1% formic acid). MS data was acquired in data-dependent fashion using a Top15 method. The exact parameters for chromatography and MS instrument settings can be retrieved from the raw files available at ProteomeXchange (PXD040024).

MS data files from single-shot experiments were processed with MaxQuant (version 2.0.1.0) using a non-redundant *Xenopus laevis* data base available at ProteomExchange (PXD040024)[45]. Raw data were normalized using the label-free quantitation (LFQ) algorithm implemented in MaxQuant. MS Data with Perseus (version 1.6.15.0)[80]. Protein groups were filtered to eliminate contaminants, reverse hits, and proteins identified by site only. For the heat map (Fig. 6a) LFQ intensities were log2 transformed and for each protein z-scores were calculated across all replicates ($N = 4$) of all four conditions. Subsequently, the average of the z-scores was calculated for each condition and selected proteins were plotted (see supplementary data 2 for the z-score of all proteins). Proteins were manually annotated and sorted according to their function in DNA replication and DNA repair. To identify proteins with significant abundance changes between the four conditions, LFQ intensities were log2 transformed and missing values were imputed with random values drawn from a normal distribution centred around the detection limit of the MS instrument (Perseus imputation; width = 0.3, down shift = 1.8). Two sample Student's t-tests were carried out in Perseus. Student's t-tests were carried out in Perseus. For these tests three valid values in at least one quadruplicate of either of the tested conditions was required. FDR was adjusted for multiple testing by the Benjamini–Hochberg procedure using a significance threshold of FDR<0.05 (see supplementary data 2). Data visualisation was carried out in R. All scripts are available upon request.

## Determination of TopBP1-proximal proteins by APEX2 biotinylation

**Sample preparation.** Proximity labelling was performed in isogenic stable 293 Flip-In (Thermo Fisher; R75007) cells stably expressing TopBP1 N-terminal tagged with APEX2 or in 293T cells transiently expressing either N-terminal or C-terminally APEX2-tagged TopBP1. 24 h before transfection, $0.8 \times 10^6$ cells were seeded onto a 6-cm plate. For each condition, four dishes were separately transfected (biological replicates). One day after transfection, cells were incubated for 30 min with 500 μM biotin phenol at 37 °C, before incubation for 1 min at RT with or without (controls) 1 mM $H_2O_2$. The medium was discarded, and cells were washed three times with quenching buffer (10 mM sodium azide, 10 mM sodium ascorbate and 5 mM Trolox in Dulbecco's PBS (Life Technologies 14190169). Cells were rinsed off the dishes with DPBS, transferred into reaction tubes before washing again with DPBS. Cells were lysed at 95 °C for 10 min in TSD buffer (50 mM Tris pH 7.5, 5 mM DTT and 1% SDS). Samples were diluted ten times with TNN buffer (20 mM Tris pH 7.9, 200 mM NaCl and 0.5% NP-40 alternative) and benzonase (75 U per sample) (Sigma E1014). Samples were incubated for 15 min at 4 °C and centrifuged at $21,000 \times g$ for 2 min at 4 °C. Supernatants were incubated over night with 15 μl streptavidin sepharose beads (Sigma GE17-5113-01) at 4 °C. Beads were washed once with TNN buffer + 0.1% SDS and twice with 25 mM ammonium bicarbonate before processing for mass spectrometry. For this, captured proteins were washed two times with $H_2O$ prior to on-bead digestion to remove MS incompatible buffer components. The beads were taken

up in 100 μl 0.8 M urea, 50 mM ammonium bicarbonate buffer (ABC) and supplemented with 5 mM DTT. After incubation at 37 °C for 30 min, 10 mM iodoacetamide (IAM) was added and incubated for 30 min at room temperature in the dark while shaking at 1500 rpm (Thermomixer C, Eppendorf). The IAM was quenched with 11 mM DTT. Trypsin was added to a total of 300 ng Trypsin per sample. The samples were incubated over night at 37 °C, shaking at 1150 rpm before stopping by 1% (vol/vol) formic acid (FA). After bead collection by centrifugation (650 × g, 5 min) 100 μl supernatant were transferred to a Eppendorf Lo-bind 1.5 ml tube. The remaining beads were incubated with 50 μl 1% formic acid for (1150 rpm, 5 min rpm, RT) and after collecting the beads once more by centrifugation (650 × g, 5 min) the supernatant was combined with the first one. The sepharose beads were discarded. To remove residual sepharose beads, the combined peptide containing solutions were passed over pre-equilibrated (50 μl 0.5% formic acid) home-made 2-disc Glass microfiber StageTip (disc material: GE Healthcare; pore size: 1.2 μM; thickness: 0.26 mm; 50 × g, 2 min). The cleared tryptic digests were then desalted on home-made C18 StageTips as described[81]. Briefly, peptides were immobilized and washed on a 2 disc C18 (Empore) StageTip. After elution from the StageTips, samples were dried using a vacuum concentrator (Eppendorf) and the peptides were taken up in 0.1% formic acid solution (10–15 μl) and directly used for LC-MS/MS experiments.

**LC-MS/MS settings.** MS Experiments were performed on an Orbitrap Fusion LUMOS instrument (Thermo) coupled to an EASY-nLC 1200 ultra-performance liquid chromatography (UPLC) system (Thermo). The UPLC was operated in the one-column mode. The analytical column was a fused silica capillary (75 μm × 46 cm) with an integrated fritted emitter (CoAnn Technologies) packed in-house with Kinetex 1.7 μm C18-XB core shell beads (Phenomenex). The analytical column was encased by a column oven (Sonation PRSO-V2) and attached to a nanospray flex ion source (Thermo). The column oven temperature was set to 50 °C during sample loading and data acquisition. The LC was equipped with two mobile phases: solvent A (0.2% FA, 2% Acetonitrile, ACN, 97.8% H₂O) and solvent B (0.2% FA, 80% ACN, 19.8% H₂O). All solvents were of UPLC grade (Honeywell). Peptides were directly loaded onto the analytical column with a maximum flow rate that would not exceed the set pressure limit of 980 bar (usually around 0.4–0.6 μl/min). Peptides were subsequently separated on the analytical column by running a 105 min gradient of solvent A and solvent B (start with 3% B; gradient 3% to 9% B for 6:30 min; gradient 9% to 30% B for 62:30 min; gradient 30% to 50% B for 24:00 min; 50% to 100% B for 2:30 min; 100% for 9:30 min) at a flow rate of 300 nl/min. The mass spectrometer was controlled by the Orbitrap Fusion Lumos Tune Application (version 3.3.2782.28) and operated using the Xcalibur software (version 4.3.73.11). The mass spectrometer was set in the positive ion mode. The ionization potential (spray voltage) was set to 2.5 kV. Source fragmentation was turned off. Precursor ion scanning was performed in the Orbitrap analyser (FT; Fourier transform mass spectrometer) in the scan range of m/z 370–1500 and at a resolution of 240,000 with the internal lock mass option turned on (lock mass was 445.120025 m/z, polysiloxane)[82]. AGC (automatic gain control) was set to standard and acquisition time to auto. Product ion spectra were recorded in a data-dependent fashion in the IT (IT; ion trap mass spectrometer) in a variable scan range (auto setting) and at rapid scan rate. Peptides were analysed using the setting top speed (repeating cycle of full precursor ion scan (AGC target set to 300%; acquisition time set to auto) followed by dependent MS2 scans for 3 s (minimum intensity threshold 4 × 10³)). The MS2 precursor ions were isolated using the quadrupole (isolation window 1.6 m/z) and fragmentation was achieved by Higher-energy C-trap dissociation (HCD) (normalized collision mode (stepped setting) and normalized collision energy set to 27, 32, 40%)). During MS2 data acquisition dynamic ion exclusion was set to 20 s. Only charge states between 2-7 were considered for fragmentation.

**Data processing.** RAW spectra were submitted to an Andromeda[83] search in MaxQuant (v2.0.3.0) using the default settings (Cox and Mann, 2008)[84]. Label-free quantification and match-between-runs was activated[85]. The MS/MS spectra data were searched against the Uniprot *H. sapiens* reference database (one protein per gene; UP000005640_9606_OPPG.fasta, 20585 entries, downloaded 1/10/2022) and a dedicated database containing the APEX2-myc sequence (ACE_0741_SOI_v01.fasta; 1 entry). All searches included a contaminants database search (as implemented in MaxQuant, 246 entries). The contaminants database contains known MS contaminants and was included to estimate the level of contamination. Andromeda searches allowed oxidation of methionine residues (16 Da) and acetylation of the protein N-terminus (42 Da) as dynamic modifications and the static modification of cysteine (57 Da, alkylation with iodoacetamide). Enzyme specificity was set to Trypsin/P with two missed cleavages allowed. The instrument type in Andromeda searches was set to Orbitrap and the precursor mass tolerance was set to ± 20 ppm (first search) and ± 4.5 ppm (main search). The MS/MS match tolerance was set to ±0.5 Da. The peptide spectrum match FDR and the protein FDR were set to 0.01 (based on target-decoy approach). Minimum peptide length was 7 aa. For protein quantification unique and razor peptides were allowed. Modified peptides were allowed for quantification. The minimum score for modified peptides was 40. Label-free protein quantification was switched on, and unique and razor peptides were considered for quantification with a minimum ratio count of 2. Retention times were recalibrated based on the built-in nonlinear time-rescaling algorithm. MS/MS identifications were transferred between LC-MS/MS runs with the match-between-runs option in which the maximal match time window was set to 0.7 min and the alignment time window set to 20 min. The quantification is based on the value at maximum of the extracted ion current. At least two quantitation events were required for a quantifiable protein. Further analysis and filtering of the results was done in Perseus v1.6.10.0.[80]. For quantification, we combined related biological replicates to categorical groups and investigated only those proteins that were found in at least one categorical group in a minimum of 3 out of 4 biological replicates. Comparison of protein group quantities (relative quantification) between different MS runs is based solely on the LFQ's as calculated by MaxQuant, MaxLFQ algorithm[85]. Statistical evaluation was done by two-sided Student's t-testing (FDR = 0.05, S0 = 0.1)

**Analysis of the TopBP1-GINS by cross-linking mass spectrometry**
**Sample preparation.** 50 μg (1.15 μg/μl) of each GINS and TopBP1-BRCT0-5-strep (amino acids 1–766) were incubated for 45 min on ice in cross-linking buffer (20 mM HEPES pH 8.0, 150 mM NaCl, 0.01% Tween-20, 2% Glycerol, 0.5 mM TCEP). 600 μM or 2500 μM of BS³ cross-linker were added and incubated for 30 min at 35 °C shaking. The reaction was stopped by adding ammonium bicarbonate to a final concentration of 100 mM. 90% of the cross-linked sample was used for mass spectrometry analysis of cross-links. 10% of the reaction were separated by SDS-PAGE. The coomassie-stained gel was cut in slices above the molecular weight of monomeric TopBP1-BRCT0-5-strep for subsequent mass spectrometry.

**Sample processing for mass spectrometry.** Sample preparation of cross-linked samples for LC/MS/MS is based on the SP3 protocol[86]. 30 μg of protein from each cross-linking sample was taken up in 100 μl 1× SP3 lysis buffer (final concentrations: 5% (wt/vol) SDS; 10 mM TCEP; 200 μl 40 mM chloracetamide; 200 mM HEPES pH 8) and heated for 5 min at 90 °C. After cooling the samples to room temperature (on ice) a mix of 150 μg hydrophobic (#65152105050250) and 150 μg

hydrophilic (#45152105050250) SeraMag Speed Beads (Cytiva) was added (bead to protein ratio 10 to 1) and gently mixed. Then 100 μL 100% vol/vol Ethanol (EtOH) was added before incubation for 20 min at 24 °C shaking vigorously. The beads were collected on a magnet and the supernatant aspirated. The beads were then washed 4 times with 180 μL 80% EtOH (collection time on the magnet minimum of 4 min). The beads were then finally taken up in 100 μl 25 mM ammoniumbicarbonate (ABC) containing 1 μg Trypsin (Protein:Trypsin ratio 30:1). To help bead dissociation, samples were incubated for 5 min in a sonification bath (preheated to 37 °C). Samples were incubated over night shaking at vigorously. Samples were acidified with formic acid (FA, final 1% vol/vol) before collection on a magnet. The supernatants were transferred to a fresh Eppendorf tube, before removing trace beads using a magnet for 5 min. The tryptic digests were then desalted on home-made C18 StageTips as described[81]. Briefly, peptides were immobilized and washed on a 2 disc C18 StageTip. Samples were then dried using a vacuum concentrator (Eppendorf) and the peptides were taken up in 0.1% formic acid solution (10 µl) and directly used for LC-MS/MS experiments.

**LC-MS/MS settings cross-linking mass spectrometry.** MS Experiments were performed on an Orbitrap Fusion LUMOS instrument (Thermo) coupled to an EASY-nLC 1200 ultra-performance liquid chromatography (UPLC) system (Thermo). The UPLC was operated in the one-column mode. The analytical column was a fused silica capillary (75 μm × 41 cm) with an integrated fritted emitter (CoAnn Technologies) packed in-house with Kinetex 1.7 μm C18-XB core shell beads (Phenomenex). The analytical column was encased by a column oven (Sonation PRSO-V2) and attached to a nanospray flex ion source (Thermo). The column oven temperature was set to 50 °C during sample loading and data acquisition. The LC was equipped with two mobile phases: solvent A (0.2% FA, 2% Acetonitrile, ACN, 97.8% $H_2O$) and solvent B (0.2% FA, 80% ACN, 19.8% $H_2O$). All solvents were of UPLC grade (Honeywell). Peptides were directly loaded onto the analytical column with a maximum flow rate that would not exceed the set pressure limit of 980 bar (usually around 0.4–0.6 μL/min). Peptides were subsequently separated on the analytical column by running a 70 min gradient of solvent A and solvent B (start with 2% B; gradient 2% to 6% B for 5:00 min; gradient 6% to 25% B for 42:00 min; gradient 25% to 40% B for 15:00 min; 40% to 98% B for 1:00 min; 98% for 7:00 min) at a flow rate of 350 nl/min. The mass spectrometer was controlled by the Orbitrap Fusion Lumos Tune Application (version 3.3.2782.28) and operated using the Xcalibur software (version 4.3.73.11). The mass spectrometer was set in the positive ion mode. The ionization potential (spray voltage) was set to 2.5 kV. Source fragmentation was turned off. Precursor ion scanning was performed in the Orbitrap analyser (FT; Fourier transform mass spectrometer) in the scan range of $m/z$ 370–1600 and at a resolution of 120,000 with the internal lock mass option turned on (lock mass was 445.120025 $m/z$, polysiloxane)[82]. AGC (automatic gain control) was set to standard and acquisition time to auto. Product ion spectra were recorded in a data-dependent fashion in the FT in a variable scan range (auto setting) and at 15,000 resolution. Peptides were analysed using a top speed regime (repeating cycle of full precursor ion scan (AGC target set to standard; acquisition time 200 ms) followed by dependent MS2 scans for 5 s (minimum intensity threshold $2 \times 10^4$). The MS2 precursor ions were isolated using the quadrupole (isolation window 2.0 $m/z$) and fragmentation was achieved by Higher-energy C-trap dissociation (HCD) (normalized collision mode set to stepped and normalized collision energy set to 27, 30, 33%). During MS2 data acquisition dynamic ion exclusion was set to 60 s. Only charge states between 3–7 were considered for fragmentation.

**Data processing protocol.** The searches for cross-linked peptides were performed with two different search engines: MetaMorpheus

(MM) version 0.0.320[87] and pLink2 version v2.3.9[88]. Searches were performed on the Thermo RAW files using the database ACE_0704_SOI_plus_con_v01.fasta (244 entries). The database contains the sequences for TopBP1 (1–766), PsfI, PsfII, PsfIII, Sld5 and 239 known contaminating proteins found in MS samples. The peptide spectrum match FDR for MM was 0.01 and for pLink2 0.05 (based on target-decoy approach, decoys are generated by the software). The settings for MetaMorpheus were: cross-linker name = DSS _KSTY (note: DSS and $BS^3$ have identical cross-linker size); cross-linker type = non-cleavable; cross-linker mass = 138.06808; cross-linker modification site 1 = K; cross-linker modification site 2 = KSTY; protease = trypsin; maximum missed cleavages = 3; minimum peptide length = 5; maximum peptide length = 60; initiator methionine behaviour = Variable; max modification isoforms = 1024; fixed modifications = Carbamidomethyl on C, variable modifications = Oxidation on M; parent mass tolerance(s) = ±10 ppm; product mass tolerance = ±20 ppm. The settings for pLink2 were: cross-linker name = DSS-BS³-KSTY; cross-linker type = non-cleavable; cross-linker mass = 138.06808; cross-linker modification site 1 = K; cross-linker modification site 2 = KSTY; protease = trypsin; maximum missed cleavages = 3; minimum peptide length = 6; maximum peptide length = 60; fixed modifications = Carbamidomethyl on C, variable modifications = Oxidation on M; parent mass tolerance(s) = ±20 ppm; product mass tolerance = ±20 ppm. The results from both searches were converted to the ProXL XML format using the respective converters (metaMorph2ProxlXML.jar, plink2toProxlXML.jar, follow links on the ProXL website; https://proxl-ms.org/) and uploaded to the ProXL server[89]. Analysis and evaluation of cross-links was performed on our local ProXL Server. The results from both searches was analysed together.

## Computational modelling

Locally implemented AlphaFold2-Multimer version 2.2.0[90] was used to predict the binding interfaces of GINS and TopBP1 using the full-length sequences of each GINS subunit and residues 301–766 of human TopBP1. 5 models were obtained, using 5 seeds per model, by default and ranked with the combined score of ipTM+pTM implemented in AlphaFold2-Multimer[90,91]. The top ranked model used for figure production and statistical plots were produced using Alphapickle 1.5.4 for the top seed from each model.

Blind molecular docking for insight into the GINI binding region used the GINI sequence DEDLLSQY of TopBP1 (residues 487–494) in the servers Cluspro[92] (accessed on 13/06/2022), HPEPDOCK[93] (accessed on 13/06/2022), and MDockPeP[94–96] (accessed on 13/06/2022) web servers with 200 solutions per server. We further extended, refined, and re-scored the sampling, by generating 10 additional conformations per web server solution (for a total of 6000 solutions) with FlexPepDock[97]. The top-10 best-scored solutions were selected for analysis and visualization.

The binding free energies of GINS with BRCT4 and GINI were estimated based on the best scored AlphaFold2 model using the web servers PRODIGY[98] and PPI-Affinity[99]. The region of GINI employed for the estimation comprised residues 481 to 496. Both PRODIGY and PPI-Affinity analyse the residues in the binding interface for free energy estimation. PRODIGY uses a linear amino acid chain model, whilst PPI-Affinity uses a structure-based machine learning algorithm.

## Cryo-electron microscopy

**Grid preparation.** Quantifoil 1.2/1.3, 300 mesh copper grids (Quantifoil) were glow discharged using a Tergeo Plasma Cleaner (Pie Scientific) with an indirect plasma treatment for 30 s. Grids were loaded into a Leica EM GP2 (Leica microsystems) and 3 µl of peak fractions from size exclusion chromatography were diluted to 0.1 mg/ml and applied to the front of the grid, with an additional 0.5 µl buffer applied to the grids rear, before back blotting for 3 s and plunging into an ethane propane mix.

**Electron microscopy and data processing.** For dataset 1, grids were stored in liquid nitrogen prior to imaging at 300 kV on a FEI Titan Krios (Thermo Fisher Scientific) equipped with K3 detector (Gatan). 18,945 movies were collected, using data acquisition software EPU (Thermo Fisher Scientific), at a magnification of 105,000 and a pixel size of 0.85 Å using a total dose of 50 e-/Å$^2$. These were motion corrected in 5 × 5 patches using MOTIONCOR2[100] implemented in RELION4.0[44] before importing into cryoSPARC[43]. Micrographs were CTF corrected prior to blob picking and extraction of an initial set of particles. Subsequent filtering by 2D classification removed most of these leaving a cleaned set of particles that were used to produce 4 initial models. The 2D classes and initial models appeared to have at least two distinct sub populations, one containing just the GINS complex alone and one that also had some additional density. In order to distinguish between these two sets of particles heterorefinement was performed to split the two groups. The first population of particles containing only the GINS complex were then imported into RELION4.0 before several cycles of 3D classification, to yield a final set of 111,455 particles that gave a volume at 5.9 Å resolution used to produce the GINS alone volume used in (Supplementary Fig. 16a). The second group of TOPBP1-GINS particles were used to train a TOPAZ model before implementing further picking using TOPAZ[101]. The resulting particles were again filtered by rounds of 2D classification and heterorefinement before exporting the particles, that appeared to consist of the full TOPBP1-GINS complex, into RELION4.0. Several cycles of 3D classification were then performed with classes appearing to lack the additional density eliminated, to yield a final set of 208,115 particles. 3D refinement and post-processing of these gave a volume at 4.71 Å resolution into which previously solved crystal structures of the GINS complex (29EX) and the central BRCT4/5 module of TopBP1 (3UEN) could be docked.

For dataset 2, grids were stored in liquid nitrogen prior to imaging at 300 kV on a FEI Titan Krios (Thermo Fisher Scientific) equipped with Fakcon 4i detector (Thermo Fisher Scientific). 8808 movies were collected, using data acquisition software EPU (Thermo Fisher Scientific), at a magnification of 120,000 and a pixel size of 0.74 Å using a total dose of 39.69 e-/Å$^2$. The same processing schedule was followed as for dataset one to yield a final set of 154,278 particles. 3D refinement and post-processing of these gave a volume at 4.1 Å resolution into which the previously solved model could be docked alongside the predicted GINI helix from the AlphaFold2-Multimer model. Minor adjustments to the model were made in Coot before refinement of the final model using PHENIX[102].

### Data presentation
Figure panels for all structural data were produced using ChimeraX-1.3.

### Reporting summary
Further information on research design is available in the Nature Portfolio Reporting Summary linked to this article.

## Data availability
The raw mass spectrometry proteomics data for the proximity-dependent biotinylation experiments, the cross-linking experiments and the chromatin mass spectrometry (CHROMASS) experiments generated in this study, have been deposited to the ProteomeXchange Consortium via the PRIDE[103] partner repository (https://www.ebi.ac.uk/pride/archive/) with the dataset identifiers PXD040000 (PDB experiments), PXD040156 (CL-MS experiments) and PXD040024 (CHROMASS experiments). Cryo-EM maps and refined coordinates generated in this study have been deposited in the EMDB and Protein Databank with the identifiers EMD-16916 and PDB ID 8OK2, respectively. Published Protein Database entries used in this study: 2E9X (GINS crystal structure); 7PFO (core human replisome); 3UEN (TopBP1-BRCT4/5 crystal structure); 6XTX (human DNA-bound CMG). Source data are provided with this paper.

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

## Acknowledgements

We thank Nora Cronin, Rebecca Thompson and Daniel Maskell, Yehuda Halfon, Pascale Schelleberger and Fabienne Beuron for assistance with cryo-EM. We also thank Svenja Heimann and Jenny Bormann for technical assistance with cross-linking and APEX2 mass spectrometry and Ken L. Dreger for our local ProXL server. We further thank Stefan Westermann, Hemmo Meyer and Olaf Stemmann for sharing reagents and expertise on biochemistry and *Xenopus* egg extracts. The lab of D.B. was supported by the SFB1430 (Project-ID 424228829, FOR2800) and RTG1739 of the Deutsche Forschungsgemeinschaft (DFG, German Rersearch Foundation) and the NRW Rückkehrerförderprogramm (state of North-Rhine-Westphalia), the labs of A.W.O and L.H.P. by Cancer Research UK Programme Grants C302/A14532 and C302/A24386, and the lab of M.R. by the FOR2800 research group (DFG). E.S.G. acknowledges funding from the DFG under Germany's Excellence Strategy (EXC 2033 – 390677874 – RESOLV), the instrumentation programme of the

DFG (436586093) and CRC1430 (Project-ID 424228829). M.K. and F.K. are grateful for funding by the DFG (DFG, INST 20876/322-1 FUGG and CRC1430).

## Author contributions

M.D. performed cryo-EM and biochemical experiments with assistance by I.A.B. B.T. performed biochemical binding studies by pulldown, most mutation analyses, some *Xenopus* egg extract experiments, and Colab Alphafold predictions that identified the GINI binding site in GINS. M.P. performed experiments using *Xenopus* egg extracts and proximity biotinylation experiments. A.M. generated many point mutant cDNAs and assisted in experiments by B.T. and M.P. Y.H.A. and E.S.G. performed computational modelling and Alphofol2-multimer predictions and analyses. H.S. performed and analysed cross-linking-mass spectrometry under supervision of B.T. and including mass spectrometry by F.K. and M.K. M.R. performed CHROMASS experiments together with M.P. F.K. and M.K. performed mass spectrometry for proximity biotinylation and cross-linking experiments. A.O.W., L.H.P., M.D. and D.B. conceptualised and supervised the project. D.B. and M.D. wrote the manuscript. D.B. coordinated the work and performed individual revision experiments.

## Funding

## Competing interests

The authors declare no competing interests.
