## [Peer Review File · Nature Communications]

TopBP1 utilises a bipartite GINS binding mode to support genome replicationEditorial Note: This manuscript has been previously reviewed at another journal that is not operating a transparent peer review scheme. This document only contains reviewer comments and rebuttal letters for versions considered at *Nature Communications* .

REVIEWER COMMENTS

Reviewer #1 (Remarks to the Author):

The authors performed a thorough revision of their work including new experiments. I believe the manuscripts should be accepted after the following items are addressed:

1. "The two active helicases to pass each other, engaging with the DNA fork with their respective N-terminal faces". Two issues. i. "to pass" should be "pass". ii. "N-terminal faces" should be "N-terminal faces first".
2. Line 76 "CMG bypass" is confusing because bypass is a concept in genetics. The authors should think of another way of phrasing this.
3. Line 62: head-to-head
4. Line 65: "converting each pre-RC into two bi-directional replisomes" – the preRC supports establishes bidirectional replication, however the 2 replisomes that form out of a single preRC themselves travel unidirectionally along the leading-strand template.
5. Line 82 (and elsewhere): RecQL4 is mentioned as potential homolog of Sld2. Recent work from the Labib, Walter, Gambus and Hashimoto laboratories establishes DONSON as a functional homolog of yeast Sld2 in metazoans. RecQL4 acts downstream of CMG formation instead. DONSON is mentioned in other passages but I think it should replace RecQL4 here, in my opinion.
6. Line 126: "Our data provide a model and framework for future investigation of CMG formation through GINS and Cdc45 loading by TopBP1 and the Treslin-MTBP complex" This manuscript does not provide as much novel insight into Cdc45 loading by the Treslin-MTBP complex.
7. Line 362 "Recent insight into origin firing has largely come from studies focussed on understanding how the yeast Mcm2-7 helicase operates as a molecular machine to build replication forks". The authors should update this statement including the recent structure of a double CMG bound by a DONSON dimer, obtained from material isolated from replicating chromatin, established with the *Xenopus* egg extract (Cvetkovic et al Mol Cell 2023).
8. line 387 "thorough CDK-mediated binding" I think this should be through?
9. Lines 464-466: Dimerisation of Sld7 and MTBP is discussed. The authors could also mention DONSON homodimerisation.
10. Figure 6a: Why is there a "?" next to Cdt1 leaving the DH. Also include Cdc6 dissociation. Also, I think it is appropriate to change Ticcr in the table and the cartoons to Treslin, as it is only referred to as Treslin throughout the manuscript.
11. Figure S4: Please provide panel identifiers and corresponding legends for this Figure, similar to Figure S11.

Reviewer #2 (Remarks to the Author):

The revised manuscript from Day, Tetik, Parlak et al has significantly improved the overall quality of the manuscript, which already contained a lot of good work. However, my main concern about the manuscript still stands – overall it is good work but I am not convinced that it represents much of an advance. I'm also concerned that the proposed role for TOPBP1 BRCT4 does not fit the data. I think the manuscript would certainly be suitable for publication if the conclusions were realigned to fit the data, but even then only in a more specialized journal. My main remaining concerns are summarized below:

1. I agree with the authors that they are the first to identify the interaction between TOPBP1-BRCT4 and GINS and characterize this protein-protein interaction. The problem is that they do not put forward a clear model for this interaction that is supported by data. Fig 5F shows that TOPBP1-BRCT4 supports DNA synthesis in *Xenopus* egg extracts when the GINI domain of TOPBP1 is also absent. This data shows that the BRCT4 domain is important for either replication initiation or elongation, which together are responsible for the total DNA synthesis measured in this assay. Figs 6D and S12B Show that TOPBP1 BRCT4 has no effect on the abundance of replisome (CMG) components on DNA, in either the presence or absence of the GINI domain. This data shows that TOPBP1 BRCT4 is not important for replisome assembly/origin firing/initiation, which is responsible for the abundance of replisome components in this assay. (Termination, i.e. replisome unloading, cannot have an affect because it is blocked by aphidicolin addition). To put it another way: an effect is observed for BRCT4 when initiation and elongation are measured (Fig 5F) but not when initiation only is measured (Figs 6D, S12B). Together, these data indicate that the TOPBP1-BRCT4 domain is important for replication elongation and not replisome assembly/origin firing/initiation. In contrast, the authors propose a role for TOPBP1-BRCT4 in origin firing (lines 40-42 in the abstract), which does not fit the data in my opinion.
2. The structural work is good information but I don't see how the structure itself adds much. The authors point to the predictive power of the structure based on the fact that the BRCT4-GINS interaction was identified. However, AlphaFold Multimer also clearly predicts the same protein-protein interaction and it's not clear to me the authors' structure adds much beyond this.
3. Similarly to point 2, I think the proteomics is good work but I honestly don't see how it advances our understanding at all. The authors also did not do a good job of addressing this point in their rebuttal.
4. The coordination between TOPBP1 ejection and Pol epsilon arrival is interesting but still speculative.

Reviewer #3 (Remarks to the Author):

In their revised manuscript Day et al. have addressed many of the comments from the initial reviews and improved the work in several important ways. Overall, the insights gained are interesting and are a nice complement to recent studies on DONSON and the assembly of the CMG via the pre-LC complex (e.g. Lim et al. 2023 which also modelled the TOPBP1-GINS interactions with AlphaFold). However, several minor concerns remain, primarily with the presentation of the data.

1. Fig. 2a and S4 – The quoted resolution of 4.6 Å for the GINS-TopBP1-BRCT4/5 construct appears to be an overestimation, perhaps related to the problems of preferred orientation highlighted by reviewer 1. In the current figures, there is no evidence of clearly defined alpha-helices as would be expected for a 4.6 Å resolution.
2. Fig. 4b and S11 – From the figures presented, it remains very challenging to assess the quality of

the model fit to the map and therefore the resolution of the map.

3. Line 285 – ‘showed moderate reductions’ – quantification of the data would be more meaningful.

4. Sup. Fig 15 a,b – Missing control. The same results could be obtained if TopBP1 interacted non-specifically with the resin.

5. Given that, within the CMG, PolE2 and the BRCT4 domain are predicted to compete for binding (not PolE2 and GINI), a competition pulldown assay between the TopBP1-BRCT0-5-Gcc (dependent on BRCT4 for GINS interaction) and PolE2N against bead immobilized GINS may be more informative. If PolE2N efficiently competes with the BRCT4 domain interaction but cannot out-compete TopBP1-GINS binding in the presence of the GINI domain, this would suggest that the Polymerase epsilon and TopBP1 interactions are not necessarily mutually exclusive within the CMG context either (line 334).

6. Sup. Fig 16 a. GINS – BRCT4 interaction labelled as an essential interaction.

7. In places, the article remains difficult to read and could be significantly improved.

Examples/suggestions:

a. In the introduction, remove unnecessary details. For example, discussing the N-terminal domains of the MCMs in an MCM-DH, the ejection of the lagging strand or ADP-ATP exchange are not relevant for understanding the work described.

b. Also consider replacing the use of pre-RCs (which are currently poorly defined) and instead describe MCM double hexamers (MCM-DHs), which are more specific and clearly distinct from Pre-LCs.

c. Line 286 – ‘consistent with the observed reduced affinity of BRCT4 for GINS’ – I believe that this is trying to say ‘consistent with the observed reduced interaction between TopBP1 and GINS in the absence of a functional GINI domain’ (or something similar)?

d. Line 300-303 – geminin is introduced as a control without any explanation of what it is or why it is added, which will make interpretation of the data difficult for a non-expert. ‘when buffer was added’ is vague and unclear when the relevant information is that TopBP1 has been depleted from the extract without add-back.

e. Line 318-320 and line 425-428 – It is unclear to me what these sentences are proposing in terms of ‘aggravating’ the defects of the GINI mutants.

f. Line 325 – ‘We find evidence that, in pre-LC’ – figure reference is missing. The next figure reference (7a-c) refers to interactions in the context of the CMG (not pre-LC), which is discussed later in the text (line 332).

g. Line 334-335 – ‘binding of DNA Pol epsilon and TopBP1 seem mutually exclusive’ – I do not agree, as the GINI interaction motif (the most important interaction site) does not appear to be in competition with PolE2. Moreover, the crosslinking data in Fig. S10 appears to support the conclusion that TopBP1 can interact with GINS via the GINI domain in the absence of interactions between BRCT4 and GINS. This does not mean that Pol epsilon isn’t involved in TopBP1 turnover from GINS bound the CMG, but this hypothesis has not been tested.

8. The figure legends could also still be improved substantially. Whilst some are detailed, others contain little to no information. Mostly, it should be made clear what the reader is being shown, so that the figures can be interpreted in isolation as much as possible. Examples/suggestions:

a. Fig. 4a, the image presumably shows a transparent surface representation for the AlphaFold model but given the similarity to other figures showing EM density maps, this could be misleading.

b. Fig. 5a. Line 1438 – ‘of’ typo; no mention of control pulldown; ‘TopBP1 strep versions’ – does this refer to TopBP1-BRCT0-5 and mutants thereof?

c. Fig. 7 – c – What are the structures we are looking at? Are they new structures from the paper, or AlphaFold models? It should be made clear that the GINS-PolE2 model is a component of the larger CMGE structure shown in panel b.

- d. Line 1472 – Figure 6 does not show the Coomassie stained histone loading control.
- e. Fig. S4 – title: 'GINS-TopBP1-BRCT4' – in the main text line 162, the TopBP1 protein used in this experiment is described as a TopBP1-BRCT4/5. In line 164 it is described as TopBP1-BRCT Δ 0/1/2. Consistent naming of the different constructs would improve readability.
- f. Fig. S11 – the legend appears to have been copied and pasted from the description of Fig. S4 and both could be more informative.

9. Other:

- a. Line 62 – 'head-two-head' typo
- b. Line 306 – Figure reference missing (Fig. 6d)
- c. Line 387 – typo 'thorough'.

28th November 2023

Dear reviewers,

Thank you very much for the valuable suggestions to improve our manuscript. We revised the manuscript accordingly. We now provide more specific insight into how the binding of PolE2-N and TopBP1-BRCT4 relate to each other, and find partial competition between them. We also changed the main text and figure legends in an attempt to improve clarity and completeness of the text, and we improved display of structural data.

Please find below a point-by-point response to all issues raised.

Sincerely yours,

Point-by-point response to reviewers' concerns

Reviewer #1:

The authors performed a thorough revision of their work including new experiments. I believe the manuscripts should be accepted after the following items are addressed:

1. *"The two active helicases to pass each other, engaging with the DNA fork with their respective N-terminal faces". Two issues. i. "to pass" should be "pass". ii. "N-terminal faces" should be "N-terminal faces first".*

Done

2. *Line 76 "CMG bypass" is confusing because bypass is a concept in genetics. The authors should think of another way of phrasing this.*

The respective section has been re-written

3. *Line 62: head-to-head*

The respective section has been re-written

4. *Line 65: "converting each pre-RC into two bi-directional replisomes" – the preRC supports establishes bidirectional replication, however the 2 replisomes that form out of a single preRC themselves travel unidirectionally along the leading-strand template.*

Changed into (line 64): "...into two replisomes travelling in opposite directions"

5. *Line 82 (and elsewhere): RecQL4 is mentioned as potential homolog of Sld2. Recent work from the Labib, Walter, Gambus and Hashimoto laboratories establishes DONSON as a functional homolog of yeast Sld2 in metazoans. RecQL4 acts downstream of CMG formation inste*

ad. DONSON is mentioned in other passages but I think it should replace RecQL4 here, in my opinion.

Changed. We now name both DONSON and RecQL4 as potential Sld2 equivalents, because the functions of Sld2, RecQL4 and DONSON have not been fully understood and RecQL4 shares sequence homology with Sld2.

6. *Line 126: "Our data provide a model and framework for future investigation of CMG formation through GINS and Cdc45 loading by TopBP1 and the Treslin-MTBP complex" This manuscript does not provide as much novel insight into Cdc45 loading by the Treslin-MTBP complex.*

We prefer to leave this expression, because GINS loading in complex with TopBP1 will have to be taken into account when the roles of Treslin and MTBP will be investigated. In that sense, our work provides part of a framework important for such future research.

7. Line 362 "Recent insight into origin firing has largely come from studies focussed on understanding how the yeast Mcm2-7 helicase operates as a molecular machine to build replication forks". The authors should update this statement including the recent structure of a double CMG bound by a DONSON dimer, obtained from material isolated from replicating chromatin, established with the *Xenopus* egg extract (Cvetkovic et al Mol Cell 2023).

To make our intentions clearer clearer, we changed the text. We mean to differentiate between progress made from about 2015 using yeast and very recent progress on DONSON and pre-LC.

Line 381: "In the past decade, insight into origin firing has largely come from studies focussed on understanding how the yeast Mcm2-7 helicase operates as a molecular machine to build replication forks ^{6, 10, 11, 12, 14, 16, 17, 19, 49, 50, 51, 52}. Most recently, the aspect of how GINS and Cdc45 are loaded to activate the helicase through formation of the CMG complex has been addressed in more detail by describing the metazoan pre-LC ^{37, 38, 39, 40, 41}."

8. line 387 "thorough CDK-mediated binding" I think this should be through?

Done

9. Lines 464-466: Dimerisation of Sld7 and MTBP is discussed. The authors could also mention DONSON homodimerisation.

DONSON was added accordingly. We updated the whole section by new published results on DONSON and pre-LCs from line 462: 'After delivery of...' This is to provide a wider perspective based on recent evidence. Thank you for pointing this out.

10. Figure 6a: Why is there a "?" next to Cdt1 leaving the DH. Also include Cdc6 dissociation. Also, I think it is appropriate to change Ticcr in the table and the cartoons to Treslin, as it is only referred to as Treslin throughout the manuscript.

Done

11. Figure S4: Please provide panel identifiers and corresponding legends for this Figure, similar to Figure S11.

Done, with updated figure legends explaining each panel.

Reviewer #2

The revised manuscript from Day, Tetik, Parlak et al has significantly improved the overall quality of the manuscript, which already contained a lot of good work. However, my main concern about the manuscript still stands – overall it is good work but I am not convinced that it represents much of an advance. I'm also concerned that the proposed role for TOPBP1 BRCT4 does not fit the data. I think the manuscript would certainly be suitable for publication if the conclusions were realigned to fit the data, but even then only in a more specialized journal. My main remaining concerns are summarized below:

We thank the reviewer for remarks on the improvements to the manuscript. We interpret the data in a classical genetic way: The two sites have additive effects on replication and on origin firing. Therefore, they act in parallel pathways to support replication and origin firing. We appreciate that the degree of effects observed leave room for the conclusion that

BRCT4 may have roles during elongation that add to the replication defect. We therefore changed the title and mention this extended interpretation of the data in the text more prominently in revised manuscript version 1.

1. I agree with the authors that they are the first to identify the interaction between TOPBP1-BRCT4 and GINS and characterize this protein-protein interaction. The problem is that they do not put forward a clear model for this interaction that is supported by data. Fig 5F shows that TOPBP1-BRCT4 supports DNA synthesis in Xenopus egg extracts when the GINI domain of TOPBP1 is also absent. This data shows that the BRCT4 domain is important for either replication initiation or elongation, which together are responsible for the total DNA synthesis measured in this assay. Figs 6D and S12B Show that TOPBP1 BRCT4 has no effect on the abundance of replisome (CMG) components on DNA, in either the presence or absence of the GINI domain.

As detailed before, there is an effect of BRCT4 mutation on origin firing, albeit a weak one for reasons that we explained before. We mention the potential involvement of BRCT4 in other replication stages.

This data shows that TOPBP1 BRCT4 is not important for replisome assembly/origin firing/initiation, which is responsible for the abundance of replisome components in this assay. (Termination, i.e. replisome unloading, cannot have an effect because it is blocked by aphidicolin addition). To put it another way: an effect is observed for BRCT4 when initiation and elongation are measured (Fig 5F) but not when initiation only is measured (Figs 6D, S12B). Together, these data indicate that the TOPBP1-BRCT4 domain is important for replication elongation and not replisome assembly/origin firing/initiation. In contrast, the authors propose a role for TOPBP1-BRCT4 in origin firing (lines 40-42 in the abstract), which does not fit the data in my opinion.

See comment above.

2. The structural work is good information but I don't see how the structure itself adds much. The authors point to the predictive power of the structure based on the fact that the BRCT4-GINS interaction was identified. However, AlphaFold Multimer also clearly predicts the same protein-protein interaction and it's not clear to me the authors' structure adds much beyond this.

While we agree with the reviewer that AlphaFold2 has an awesome ability to predict protein structures, it is just that, a prediction. Currently, AF2-predicted protein complexes must be backed by independent evidence to be seriously considered. This is also what the predictomes.org database that provides pairwise prediction data on more than 200 chromatin-associated proteins emphasizes. The predicted structure of the GINS-TopBP1 complex presented in this manuscript actually illustrates one occasion where AF2 prediction may result in wrong conclusions. In the predicted AF2 structures, the B domain of Psf1 can be seen packed against the GINS complex in the position it adopts in the CMG structure (Fig4 and FigS8). In the experimental cryo-EM volume no density can be seen for the B-domain, demonstrating its flexibility (Fig2, FigS4 and FigS16).

3. Similarly to point 2, I think the proteomics is good work but I honestly don't see how it advances our understanding at all. The authors also did not do a good job of addressing this point in their rebuttal.

We are showing the data to provide rich data source accompanying the conclusions. But we agree that this does not affect the basic conclusions. We are happy to take the mass spectrometry chromatin isolation experiment out if required.

4. The coordination between TOPBP1 ejection and Pol epsilon arrival is interesting but still speculative.

We clearly state the speculative nature of this conclusion and put forward reasons underlying our speculation. We provide a more specific analysis on the partly competitive nature of Pole2 and TopBP1-BRCT4 interaction with a Psf1 region involving adjacent parts the A and B domains (Figures S15a/b).

Reviewer #3:

In their revised manuscript Day et al. have addressed many of the comments from the initial reviews and improved the work in several important ways. Overall, the insights gained are interesting and are a nice complement to recent studies on DONSON and the assembly of the CMG via the pre-LC complex (e.g. Lim et al. 2023 which also modelled the TOPBP1-GINS interactions with AlphaFold). However, several minor concerns remain, primarily with the presentation of the data.

1. Fig. 2a and S4 – The quoted resolution of 4.6 Å for the GINS-TopBP1-BRCT4/5 construct appears to be an overestimation, perhaps related to the problems of preferred orientation highlighted by reviewer 1. In the current figures, there is no evidence of clearly defined alpha-helices as would be expected for a 4.6 Å resolution.

As previously discussed, we observe that our data have a preferred orientation issue (see FigS4C), but as other views are represented, we believe the resulting volumes are still informative at the contour levels they are shown in the figures.

However, to address reviewers' concerns we have carried out a separate analysis using the Remote 3DFSC Processing Server¹, and included the output attached to this response. The histogram and directional FSC plot are included (Review Figure Panels A and B) along with a zip file containing all output files. This analysis highlights the orientation issue and anisotropy of our data, suggesting this is at least partially responsible for the reported resolution.

We can also see from local resolution estimates (Fig.S4B) that for the GINS-TopBP1 there is significant variance across several different regions of the structure, suggesting conformational flexibility.

We fully agree with the reviewer that at a resolution of 4.6 Å, we would expect to see tubes of density corresponding to alpha-helices, and as such the reported resolution is likely an overestimate. However, this resolution is based on the Fourier Shell Correlation calculated in Relion using a 0.143 cut-off, which is the standard statistical measure applied to all current cryo-EM datasets and provides a global measure of resolution.

With this said, we provide an alternative version for Figure 2, at a higher contour level where helices can clearly be seen (Reviewer Figure Panel C), especially for the BRCT domains of TopBP1. In the 'best' orientation (indicated by 3DFSC) we see clear evidence and density for alpha-helices. In the 'worst' direction, this is much less apparent. This, along with the desired inclusion of density representing other secondary structure elements informed our original choice of contour level in our figure.

We strongly believe that our maps (albeit anisotropic) allow unambiguous docking of each X-ray crystal structure, and thus perform the singular function required in our manuscript. We are also very careful to include qualifying statements such as: "Note that some ambiguity exists as to the precise molecular details due to the moderate resolution" and "TopBP1 residues Val590, Thr606 and Val610 appeared to be involved in the interface", to not allude to atomic-resolution.

We added a 'Figure for Reviewers' and a more comprehensive ZIP File with underlying data: Zip file password: 31160dd953a9ae533dce

1 - Tan, Y., Baldwin, P., Davis, J. et al. Addressing preferred specimen orientation in single-particle cryo-EM through tilting. *Nat Methods* 14, 793–796 (2017). <https://doi.org/10.1038/nmeth.4347>

2. Fig. 4b and S11 – From the figures presented, it remains very challenging to assess the quality of the model fit to the map and therefore the resolution of the map.

While we would agree that the cryoEM model is not overwhelmingly convincing in isolation, the additional density seen in the cryoEM volume is only one line of evidence that suggests the correct binding site for the GINI motif has been identified. It sits alongside the Alphafold2 modelling, the in silico docking, the cross linking mass spectrometry analysis, and the pulldown experiments using mutants for the interface.

See comment under 1. regarding resolution. At the 4.1 Angstrom resolution reported we are not looking for side chain fits. We believe the fit of the extra GINI residues (from the Alphafold2 model) into the density is visible and apparent from Fig4B. Furthermore, in the validation report section 9.3 the GINI can be seen coloured cyan indicating its inclusion in the volume.

3. Line 285 – 'showed moderate reductions' – quantification of the data would be more meaningful.

Changed accordingly from line 280: "...whereas addition of either TopBP1-BRCT0-5-Gcc or TopBP1-BRCT0-5-Gpp (BRCT4 intact) led to moderately reduced nucleotide incorporation by 21 % and 55 % (120 min time point), respectively (Fig. 5f, supplementary Fig. S13a),..."

4. Sup. Fig 15 a,b – Missing control. The same results could be obtained if TopBP1 interacted non-specifically with the resin.

Several experiments shown contain this control, among them new FigS15a and b in two separate immunoblots.

5. Given that, within the CMG, PolE2 and the BRCT4 domain are predicted to compete for binding (not PolE2 and GINI), a competition pulldown assay between the TopBP1-BRCT0-5-Gcc (dependent on BRCT4 for GINS interaction) and PolE2N against bead immobilized GINS may be more informative. If PolE2N efficiently competes with the BRCT4 domain interaction but cannot out-compete TopBP1-GINS binding in the presence of the GINI domain, this would suggest that the Polymerase epsilon and TopBP1 interactions are not necessarily mutually exclusive within the CMG context either (line 334).

We thank the reviewer for this good suggestion. We did the experiment, which sharpened our original conclusion of a partial overlap between the PolE2-N and BRCT4 binding sites on Psf1 that we now describe as part of Figs S15a/b. We want to point out though that we base our conclusion of Pole/TopBP1 exchange on steric exclusion when CMG-bound structures are compared. Competitive binding between BRCT4 and PolE2-N is only one part of the interaction between GINS and TopBP1. Pol epsilon does not overtly compete with the GINI region-dependent binding.

We originally thought that using the TopBP1-Gcc mutant in a competitive pulldown experiment is not necessary, because we had used TopBP1-WT in stringent conditions, where both GINI and BRCT4 are insufficient for a detectable GINS interaction. Consequently, we expected that competition of PolE2-N with BRCT4 should lead to lack of detectable binding. But the situation turned out to be more complex: residual affinities of both GINI and BRCT4 in stringent conditions in the presence of PolE2-N is apparently sufficient for almost normal levels of GINS binding in pulldowns, and when Psf1- Δ B is used. Using TopBP1-Gcc (new Figs S15a/b revealed A) a partial dependency of BRCT4 binding on the Psf1-B domain, and B) a clear, albeit incomplete, competitive effect of PolE2-N binding to BRCT4-GINS interaction in low-stringency conditions. Unfortunately, a considerable unspecific binding activity of MBP-PolE2-N prevented us from conclusively testing whether PolE2-N binds to GINS-Psf1- Δ B, and whether TopBP1 competes off PolE2-N from GINS.

6. *Sup. Fig 16 a. GINS – BRCT4 interaction labelled as an essential interaction.*

We now specify in legend (Fig. S17) that what is meant is that both GINS binding sites are collectively essential.

7. *In places, the article remains difficult to read and could be significantly improved.*

Examples/suggestions:

We went through the text again and made changes to focus the text (introduction) and improve readability.

a. In the introduction, remove unnecessary details. For example, discussing the N-terminal domains of the MCMs in an MCM-DH, the ejection of the lagging strand or ADP-ATP exchange are not relevant for understanding the work described.

Some detail removed

b. Also consider replacing the use of pre-RCs (which are currently poorly defined) and instead describe MCM double hexamers (MCM-DHs), which are more specific and clearly distinct from Pre-LCs.

Done. We understand pre-RCs and MCM-DH as almost synonymous terms.

c. Line 286 – ‘consistent with the observed reduced affinity of BRCT4 for GINS’ – I believe that this is trying to say ‘consistent with the observed reduced interaction between TopBP1 and GINS in the absence of a functional GINI domain’ (or something similar)?

Replaced

d. Line 300-303 – geminin is introduced as a control without any explanation of what it is or why it is added, which will make interpretation of the data difficult for a non-expert. ‘when buffer was added’ is vague and unclear when the relevant information is that TopBP1 has been depleted from the extract without add-back.

Information added in several places of this section to improve clarity.

e. Line 318-320 and line 425-428 – It is unclear to me what these sentences are proposing in terms of ‘aggravating’ the defects of the GINI mutants.

We meant that B4mut and deletion of BRCT4/5 strengthen the defect of GINI site mutants on DNA replication (nucleotide incorporation, (Figs 5f and S13).

We changed the text from line 318 accordingly:

“It is feasible that compensatory fork acceleration partially compensates for decreased replisome numbers in the Gcc mutant. Our observation that deleting BRCT4 has only minor effects on replisome formation is also consistent with the scenario that the BRCT4-GINS interaction has a role downstream of origin firing that aggravates the DNA synthesis defects (Fig. 5f, Fig. S13) of GINI site mutants”

And from line 443:

“An alternative interpretation of our TopBP1 mutant analysis seems feasible: TopBP1-D4/5 and B4mut only resulted in no DNA synthesis defect and a weak origin firing defect. This scenario is consistent with the possibility that the BRCT4-GINS interaction plays an unidentified role in replication, for example in elongation, which could contribute to the observed effect that BRCT4 inactivating mutants aggravate the defects of GINI site mutants in DNA replication assays (nucleotide incorporation).”

f. Line 325 – ‘We find evidence that, in pre-LC’ – figure reference is missing. The next figure reference (7a-c) refers to interactions in the context of the CMG (not pre-LC), which is discussed later in the text (line 332).

Changed and clarified.

g. Line 334-335 – ‘binding of DNA Pol epsilon and TopBP1 seem mutually exclusive’ – I do not agree, as the GINI interaction motif (the most important interaction site) does not appear to be in competition with PolE2. Moreover, the crosslinking data in Fig. S10 appears to support the conclusion that TopBP1 can interact with GINS via the GINI domain in the absence of interactions between BRCT4 and GINS. This does not mean that Pol epsilon isn’t involved in TopBP1 turnover from GINS bound the CMG, but this hypothesis has not been tested.

The reviewer is right that BRCT4-PolE2-N competition may not be sufficient to dissociate TopBP1 from GINS. On the reviewer’s advice, we are now providing more experiments to clarify this direct competition, which is indeed limited to BRCT4. However, we are not speaking about direct competition in the sentence mentioned, and throughout the text we explain specifically how we arrive at the conclusion that Pol epsilon and TopBP1 might

exchange. We agree that the issue remains to be experimentally closer investigated as we do not work with MCM-DH or CMG in this paper.

8. *The figure legends could also still be improved substantially. Whilst some are detailed, others contain little to no information. Mostly, it should be made clear what the reader is being shown, so that the figures can be interpreted in isolation as much as possible.*

Examples/suggestions:

We went through all figures again and made changes for clarification.

a. *Fig. 4a, the image presumably shows a transparent surface representation for the AlphaFold model but given the similarity to other figures showing EM density maps, this could be misleading.*

We agree and have adjusted the legend to make this clearer.

b. *Fig. 5a. Line 1438 – ‘of’ typo; no mention of control pulldown; ‘TopBP1 strep versions’ – does this refer to TopBP1-BRCT0-5 and mutants thereof?*

Changed

c. *Fig. 7 – c – What are the structures we are looking at? Are they new structures from the paper, or AlphaFold models? It should be made clear that the GINS-Pole2 model is a component of the larger CMGE structure shown in panel b.*

We have adjusted the legend to make this clearer.

d. *Line 1472 – Figure 6 does not show the Coomassie stained histone loading control.*

Mislabelling in the figure changed. The figure shows a Coomassie-stained gel slice.

e. *Fig. S4 – title: ‘GINS-TopBP1-BRCT4’ – in the main text line 162, the TopBP1 protein used in this experiment is described as a TopBP1-BRCT4/5. In line 164 it is described as TopBP1-BRCT0/1/2. Consistent naming of the different constructs would improve readability.*

Fixed, title adjusted to reflect it is the processing of the initial cryo-EM dataset to remove ambiguity (FigS11 title adjusted similarly).

f. *Fig. S11 – the legend appears to have been copied and pasted from the description of Fig. S4 and both could be more informative.*

Fixed, panel identifiers have been added to FigS4, with updated figure legends explaining each panel.

9. *Other:*

a. *Line 62 – ‘head-two-head’ typo*

Deleted as part of clarifying the introduction section.

b. *Line 306 – Figure reference missing (Fig. 6d)*

Figure references are more precise now in this whole section.

c. *Line 387 – typo ‘thorough’.*

Changed.

a

Sphericity = 0.788 out of 1. Global resolution = 4.35 \AA .

b

c

REVIEWERS' COMMENTS

Reviewer #3 (Remarks to the Author):

The authors have made significant improvements to the manuscript and addressed the majority of my points of concern. I believe that the revised manuscript is suitable for publication.